# Synthesis of a comprehensive population code for contextual features in the awake sensory cortex

Evan H Lyall[1,2†], Daniel P Mossing[1,2†], Scott R Pluta[2‡], Yun Wen Chu[2‡], Amir Dudai[3], Hillel Adesnik[2,4*]

[1]Biophysics Graduate Group, Berkeley, United States; [2]Department of Molecular and Cell Biology, Berkeley, United States; [3]The Edmond and Lily Safra Center for Brain Sciences and The Life Sciences Institute, The Hebrew University of Jerusalem, Jerusalem, Israel; [4]The Helen Wills Neuroscience Institute, University of California, Berkeley, Berkeley, United States

**Abstract** How cortical circuits build representations of complex objects is poorly understood. Individual neurons must integrate broadly over space, yet simultaneously obtain sharp tuning to specific global stimulus features. Groups of neurons identifying different global features must then assemble into a population that forms a comprehensive code for these global stimulus properties. Although the logic for how single neurons summate over their spatial inputs has been well explored in anesthetized animals, how large groups of neurons compose a flexible population code of higher-order features in awake animals is not known. To address this question, we probed the integration and population coding of higher-order stimuli in the somatosensory and visual cortices of awake mice using two-photon calcium imaging across cortical layers. We developed a novel tactile stimulator that allowed the precise measurement of spatial summation even in actively whisking mice. Using this system, we found a sparse but comprehensive population code for higher-order tactile features that depends on a heterogeneous and neuron-specific logic of spatial summation beyond the receptive field. Different somatosensory cortical neurons summed specific combinations of sensory inputs supra-linearly, but integrated other inputs sub-linearly, leading to selective responses to higher-order features. Visual cortical populations employed a nearly identical scheme to generate a comprehensive population code for contextual stimuli. These results suggest that a heterogeneous logic of input-specific supra-linear summation may represent a widespread cortical mechanism for the synthesis of sparse higher-order feature codes in neural populations. This may explain how the brain exploits the thalamocortical expansion of dimensionality to encode arbitrary complex features of sensory stimuli.

*For correspondence:
hadesnik@berkeley.edu

†These authors contributed equally to this work

Present address: ‡Department of Biology, Purdue University, West Lafayette, United States

**Competing interest:** The authors declare that no competing interests exist.

## Editor's evaluation

This study provides now a comprehensive and well-documented proof that in both the visual and somato-sensory cortex the response to the sum of two stimuli is not the sum of the responses to individual stimuli in a large set of neurons. This is an important piece of work that further challenges linear models of cortical function and provide useful datasets to develop alternative non-linear models.

## Introduction

To generate unique sensory percepts of complex stimuli individual cortical neurons that encode specific stimulus features must assemble into larger populations that together uniquely represent each possible stimulus. Although neurons are primarily driven by specific patterns of sensory drive within their receptive field, stimulus features beyond the receptive field potently modulate a neuron's coding properties. Understanding this modulation is key to understanding representations of complex stimuli. In some instances, stimuli that on their own drive no activity in a neuron can facilitate its firing rate when combined with stimulation in its receptive field (*Ego-Stengel et al., 2005*; *Estebanez et al., 2016*; *Hirata and Castro-Alamancos, 2008*; *Ramirez et al., 2014*; *Shimegi et al., 1999*), but far more frequently they drive suppression (*Adesnik et al., 2012*; *Angelucci et al., 2017*; *Arabzadeh et al., 2003*; *Bair et al., 2003*; *Blakemore and Tobin, 1972*; *Brumberg et al., 1996*; *Cavanaugh et al., 2002*; *Dipoppa et al., 2018*; *Ego-Stengel et al., 2005*; *Fanselow and Nicolelis, 1999*; *Gilbert, 1977*; *Kato et al., 2017*; *Mirabella et al., 2001*; *Ozeki et al., 2009*; *Shushruth et al., 2013*). In the barrel cortex, even the best contextual stimulus for a neuron typically approximates linear summation of its component inputs (*Laboy-Juárez et al., 2019*). Although suppression can improve information coding and efficiency (*Crochet et al., 2011*; *Froudarakis et al., 2014*; *Sachdev et al., 2012*; *Vinje and Gallant, 2000*), sharpen feature selectivity (*Angelucci et al., 2017*; *Jacob et al., 2008*) and enhance saliency and boundary detection (*Li, 1999*; *Li, 2009*), supra-linear summation to highly specific combinations of inputs could represent a powerful means by which cortical neurons might compute and selectively encode higher-order features of sensory stimuli. In the primary visual cortex, specific contextual stimuli have been shown to facilitate responses in a supra-linear manner, although the underlying logic is not well understood (*Levitt and Lund, 1997*; *Sillito et al., 1995*).

Despite extensive prior investigation of the logic of contextual summation on the individual neuron level (*Angelucci et al., 2017*; *Armstrong-James and Callahan, 1991*; *Bair et al., 2003*; *Blakemore and Tobin, 1972*; *Boloori and Stanley, 2006*; *Brecht et al., 2003*; *Brecht and Sakmann, 2002*; *Brumberg et al., 1996*; *Gilbert, 1977*; *Higley and Contreras, 2003*; *Moore and Nelson, 1998*; *Simons, 1985*; *Simons and Carvell, 1989*), how the brain builds a comprehensive population code that covers the high-dimensional space of global stimulus properties remains unknown. If all neurons summated their surround inputs in a similar fashion, contextual modulation would primarily act to scale activity but not generate new coding properties. If instead different neurons in a population integrated their contextual inputs uniquely, selectivity for more complex sensory stimuli could emerge across neural ensembles. Prior experiments with natural visual stimuli support the notion that cortical populations can encode highly diverse features, yet the logic of how these population codes emerge, and how heterogeneous spatial summation might support them, is essentially unknown.

Primary sensory cortical neurons massively outnumber their thalamic inputs, permitting overcomplete representations. Theoretical and experimental work has proposed that overcompleteness permits sparse representations, in which small groups of active neurons efficiently represent natural or complex inputs (*Barlow, 1961*; *de Vries et al., 2020*; *Lewicki and Sejnowski, 2000*; *Olshausen and Field, 1997*; *Stringer et al., 2019*). We tested the hypothesis that populations of cortical neurons supra-linearly summating diverse, but specific combinations of inputs could give rise to a sparse, comprehensive code for higher-order features, and potentially represent a widespread mechanism in the cortex for the synthesis of complex feature codes. We found that the diversity and specificity of this supra-linear summation yields a sparse, comprehensive population code for complex spatial features. In particular, each possible combination of whiskers or visual stimulus patches drives activity in a dedicated, similarly sized population of neurons. This mechanism may thus be critical for the ability of cortical populations to encode arbitrary complex stimuli.

## Results

### Stimulus-specific summation during active sensation in the barrel cortex

We tested the notion of diverse summation logic in both the primary somatosensory and visual cortex of the rodent. First, we took advantage of the mouse whisker system, which is naturally discretized (*Feldmeyer et al., 2013*; *Petersen, 2007*), to probe the logic of cortical integration across stimulus space in awake, actively sensing animals. We developed a novel paradigm for reliably generating active touch between a user-defined set of whiskers and a corresponding set of pneumatic actuators

(*Figure 1A*). Awake mice were head-fixed on a rotary treadmill and habituated to run at relatively high speed ( >30 cm/s), a condition in which they moved their whiskers in a rhythmic and stereotyped fashion for many minutes at a time with a protracted set point (*Pluta et al., 2015*; *Pluta et al., 2017*). All but five of the largest whiskers were completely trimmed just before the experiment, and data was collected in the next 1–2 hr prior to the onset of any trimming-induced plasticity (*Clem and Barth, 2006*; *Li et al., 2014*). Next, a set of five pneumatically controlled pistons was presented to precise locations (*Figure 1B*; *Video 1*), such that on each trial the corresponding set of the remaining 1–5 whiskers made repeated active touches (13.1±5.9 touches, mean ± s.d.) with the presented set of pistons. On each trial, the pistons entered the whisking field within one whisk cycle (~7 ms). Importantly, slight trimming of these five whiskers ensured that one and only one whisker made contact with each piston, which we confirmed using high-speed imaging (*Figure 1C*; ). A single camera with a wide depth of field was sufficient to image all five whiskers and track them independently and identify contacts of each whisker with its corresponding piston. An example trial with a video still image and whisker tracking is presented in *Figure 1C* (and see *Figure 1—figure supplement 1* and *Video 2*). In this trial pistons targeted only the B1, C1, and γ whiskers, and each whisker made 17–20 absolutely selective contacts over 25 whisk cycles with its respective piston on this trial (*Figure 1C*). Under these conditions, we sampled neural activity in the upper layers with GCaMP6s (*Figure 1D and E* see Materials and methods). Mice were not trained to respond to sensory stimulation in any way, but only data during locomotion was included, a condition when the whiskers could contact their corresponding pistons. Locomotion ensured the animals were whisking and in an alert state, although not necessarily attending to the stimulus – a feature which cannot easily be inferred from the stimulus paradigm used here.

To probe how barrel cortex neurons encoded first and higher-order tactile features, we presented all 31 possible combinations of the five pistons. Forty-eight percent (2262/4756) of identified neurons showed a significant response (see Materials and methods) to at least one of the 31 stimuli. When the pistons were presented individually to map the 'receptive field' of each touch-responsive neuron, 52% of responsive neurons exhibited a significant response to just one of the single whisker stimuli, and 12.6% responded to two or more. However, a striking 34.8% did not respond to any single whisker stimulus, but responded robustly to at least one multi-whisker combination. Note that owing to the slow kinetics of calcium measurements these responses represent the summed responses of each neuron to multiple contacts between the whiskers and the pistons over multiple whisk cycles. Moreover, since animals freely and volitionally whisked at the static piston array, the number, duration, and sequences of touches across the corresponding whiskers, as well as the bending of individual whiskers, varied within and across trials (see Whisker tracking analysis below). Thus, our data captured the average response of barrel cortex neurons to a precise set of pistons across variable sets of contacts on a cycle by cycle and trial by trial basis. It is likely the precise pattern and sequence of touches on each whisk cycle modulated the population response, but that was not captured here.

Both L4 and L2/3 excitatory neurons organized into loose spatial maps of preferred whiskers consistent with prior findings from anesthetized mice (*Clancy et al., 2015*; *Kerr et al., 2007*; *Sato et al., 2007*; *Figure 2A* and *Figure 2—figure supplement 1*). Previous work from passive stimulation paradigms in anesthetized animals revealed fine spatial structure within a barrel column (*Estebanez et al., 2016*). Although we did not find any obvious spatial clustering of cells with similar response properties, we cannot exclude it and it warrants further investigation in a large sample of animals where the barrel columns boundaries are more precisely mapped.

In many previous recordings from the barrel cortex, stimuli presented outside a neuron's receptive field typically suppressed the response to stimulation of the receptive field alone (*Arabzadeh et al., 2003*; *Brumberg et al., 1996*; *Drew and Feldman, 2007*; *Fanselow and Nicolelis, 1999*; *Laboy-Juárez et al., 2019*; *Sachdev et al., 2012*), while under certain conditions surround stimuli could facilitate responses (*Estebanez et al., 2016*; *Ramirez et al., 2014*; *Shimegi et al., 1999*). Yet all these studies employed passive stimulation of whiskers in anesthetized or sedated animals. Since active whisking is a fundamental aspect of the rodent whisker system, we asked what the logic of contextual modulation would be in awake mice, when they volitionally swept their whiskers into physical stimuli. Furthermore, we asked if most neurons in the barrel cortex shared a similar logic for summation across surround whiskers, or instead employed a heterogeneous set of rules that might facilitate a less redundant code at the population level. Unlike most of the prior studies, we found

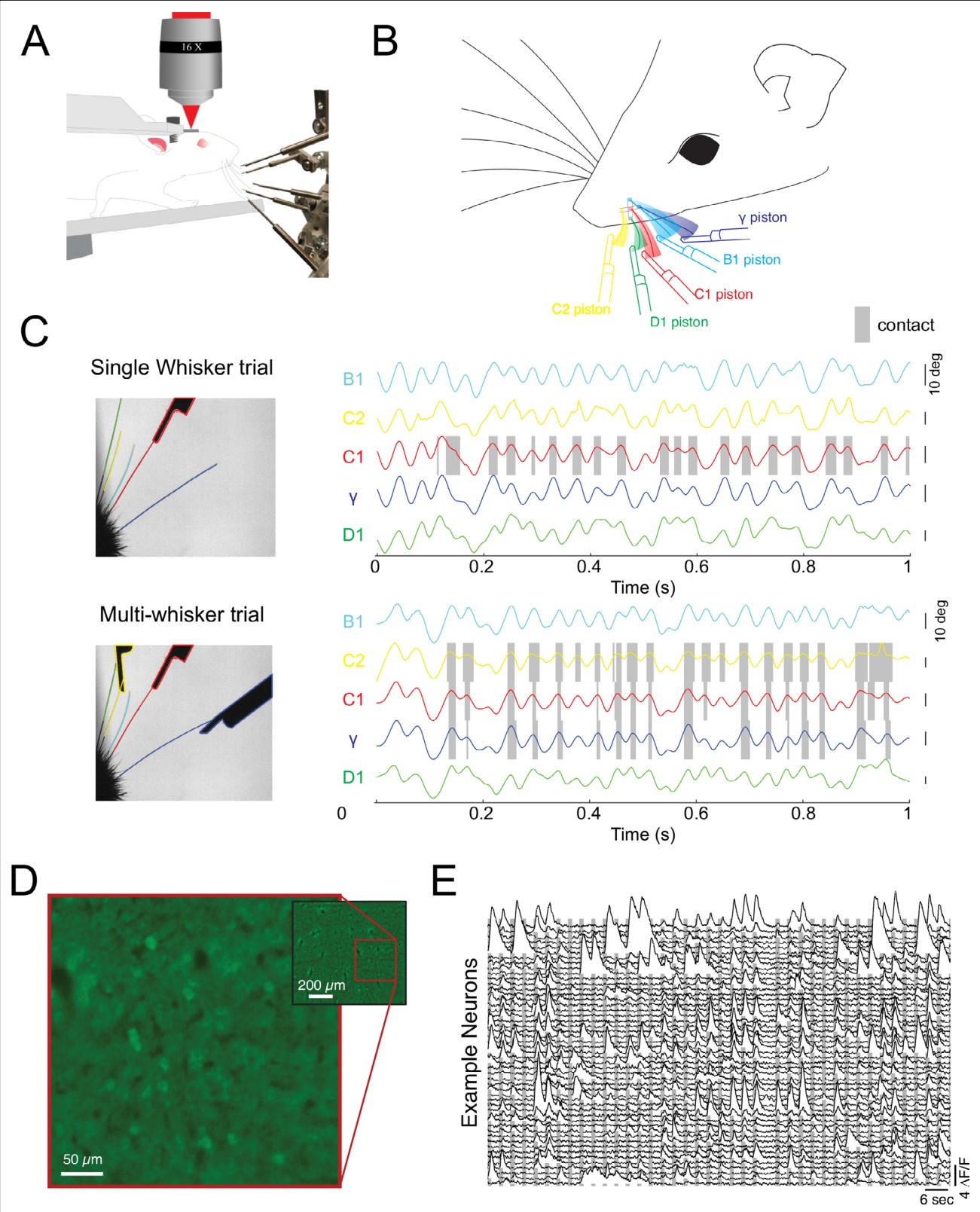

**Figure 1.** Probing cortical population codes of higher-order features during active touch. (**A**) Schematic of the experimental preparation showing a locomoting mouse under a two-photon microscope actively contacting five individual pistons with five individual whiskers (the rest are trimmed just before the experiment, see Materials and methods). (**B**) Schematic of the multi-whisker stimulator. (**C**) Top left: single frame of a high-speed video where only the C1 whisker was presented with a piston. Top right: angle traces for each of the five whiskers. Touch events between the C1 whisker and the

*Figure 1 continued on next page*

*Figure 1 continued*

C1 piston are shaded in gray. Bottom: same as above but for a trial where the C2, C1, and $\gamma$ pistons were presented. (**D**) Example field of view from a single depth (pixel values are the variance across time). (**E**) Two minutes of consecutive fluorescence measurements from 50 randomly chosen neurons in a single example experiment. Vertical gray bars indicate the presentation of different multi-piston stimuli.

The online version of this article includes the following figure supplement(s) for figure 1:

**Figure supplement 1.** Whisker tracking during an example trial.

that most neurons summed specific combinations of inputs highly supra-linearly. Consider the neuron whose calcium responses are shown in *Figure 2b*. This neuron showed no significant response to any single piston alone but exhibited the strongest activity when pistons targeting the C2, B1, D1 whiskers were presented in combination. Tuning curves for five example neurons are shown in *Figure 2c*. The first neuron showed a conventional tuning curve, exhibiting a strong and selective response to the C1 piston when presented alone, but weaker responses when any other pistons were presented in combination with the C1 piston. In contrast, neurons 2–5 (neuron three is the same as in *Figure 2b*) all showed preferred responses for different multi-piston combinations. Remarkably, for these other neurons, most other combinations of pistons had a weak or suppressive effect on the neuron's activity, giving rise to a highly selective response for their specific multi-whisker stimulus.

We next computed the difference between the actual response of each neuron to each multi-whisker stimulus, and the predicted response based on a linear sum of the responses to individual whisker stimuli ('Linear difference', *Figure 2d*). For all five neurons, most multi-whisker stimuli drove sub-linear summation (i.e. suppression) or had no effect. However, for many cells (e.g. neurons No. 2 to No. 5, *Figure 2D*), the response to the preferred multi-whisker stimulus was much greater than the linear sum of the responses to its component (single piston) stimuli. Importantly, for many of these neurons, the preferred multi-whisker response was the maximum response across the entire stimulus set.

Although calcium imaging affords the ability to measure the responses of hundreds to thousands of neurons at a time with defined spatial locations, the slow response of calcium measurements obscures any underlying higher frequency response components. To address this, we also measured multi-whisker responses using multi-electrode arrays in a different subset of mice but in very similar conditions (*Figure 2—figure supplement 4A-D*). Just as in the calcium imaging data, we found many single units that show pronounced selectivity for multi-whisker stimuli that were also generated through supra-linear summation of component inputs. For these data, we could compare the linearity of summation for individual touch responses which was not possible with calcium imaging date. These data from electrophysiology show that the core response properties we observed with calcium imaging are consistent when measuring neural activity with much greater temporal precision, although calcium imaging cannot access the fine temporal structure since individual touch-induced responses are obscured.

## A comprehensive population code for higher-order tactile stimuli in the upper layers of the barrel cortex

If many other neurons were to show similarly strong and selective responses for various

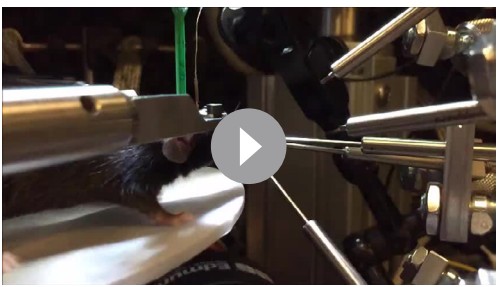

**Video 1.** Video of tactile stimulation with the novel, multi-piston tactile stimulator for head-fixed, locomoting mice.
https://elifesciences.org/articles/62687/figures#video1

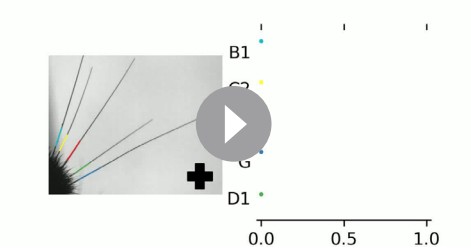

**Video 2.** High speed tracking of whiskers and contacts during tactile stimulation.
https://elifesciences.org/articles/62687/figures#video2

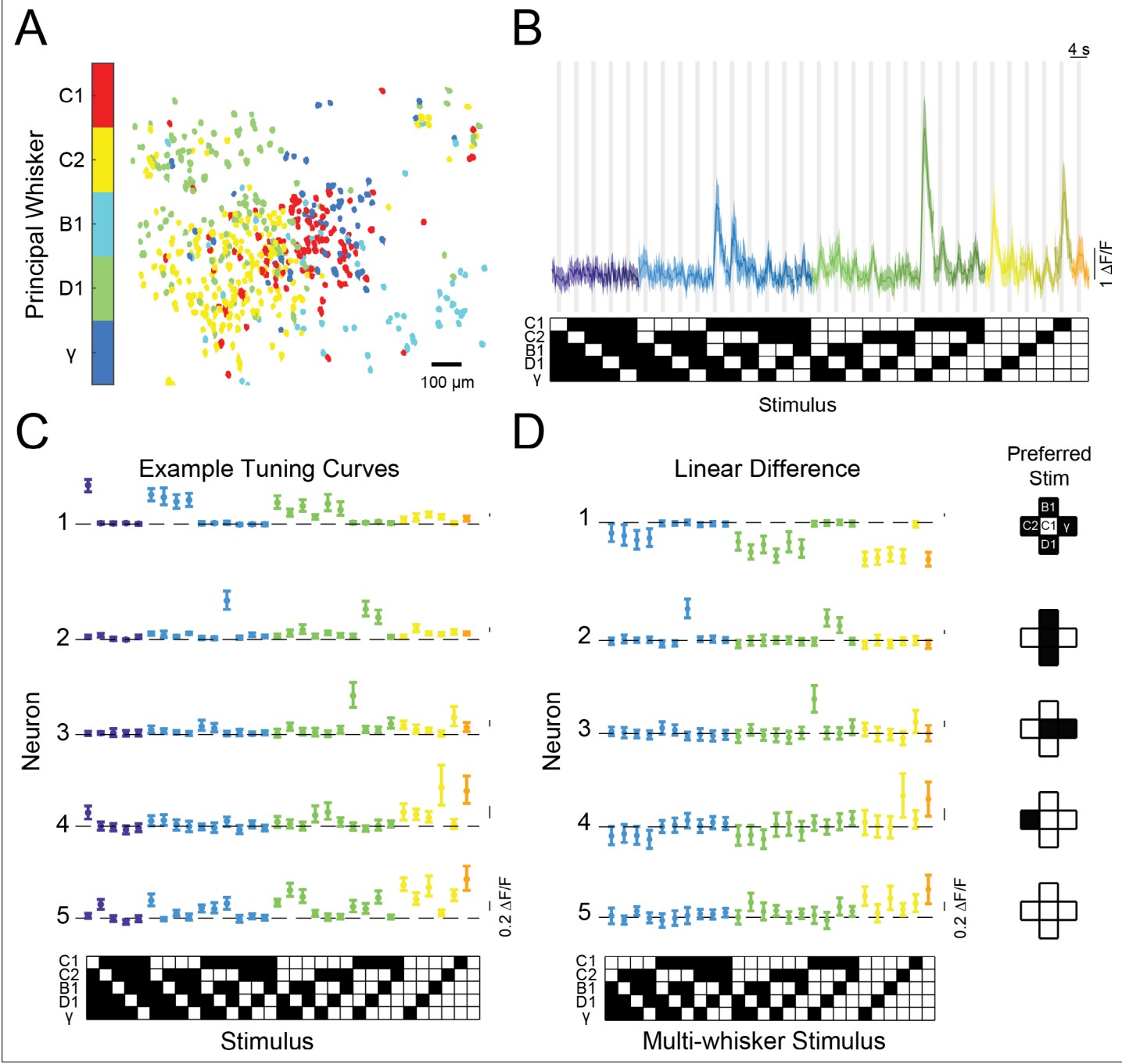

**Figure 2.** Diverse and selective representations of higher-order tactile stimuli in mouse barrel cortex. (**A**) Example map of principal whisker preference in L2/3 of one mouse. (**B**) Top: calcium responses from an example neuron to each of the 31 possible combinations of the five pistons (mean and 95% confidence intervals [C.I.]). Traces are colored-coded by the number of whiskers contacting pistons in each stimulus (1–5 whiskers). Vertical gray bars indicate presence of the tactile stimulus. Bottom: diagram of the 31 corresponding piston combination stimuli. White squares indicate the presence of the corresponding piston and black indicates the absence. (**C**) Example tuning curves for five neurons in L2/3 of S1 for all 31 stimuli and colored as in (**B**). Neuron three corresponds to the neuron shown in panel (**B**). (**D**) Left: the computed difference of the observed response to each multi-whisker stimulus from the linear sum of the component responses when each whisker contacts its corresponding piston alone for the five example neurons in (**C**), or 'Linear Difference'. All error bars are bootstrapped 95% CI. Right: diagram of each neuron's preferred stimulus.

The online version of this article includes the following figure supplement(s) for figure 2:

**Figure supplement 1.** Maps of single whisker tuning in awake, whisking mice.

**Figure supplement 2.** The transform from spiking to fluorescence in GCaMP6s-expressing neurons in *Camk2-tTa;tetO-GCaMP6s* mice.

*Figure 2 continued on next page*

*Figure 2 continued*

**Figure supplement 3.** Whisker kinematics across the stimulus set.

**Figure supplement 4.** Electrophysiological measurement of supralinear summation during multi-whisker contact.

combinations of whisker touches, with different neurons encoding all possible combinations, barrel cortex could contain a sparse, comprehensive code for higher-order tactile features in the stimulus space we probed (what has been considered an 'overcomplete' code since the number of responsive neurons in S1 exceeds the stimulus dimensionality *Olshausen and Field, 1997*). To test this possibility, we asked whether these large populations of barrel cortex neurons encoded all 31 possible piston combinations within the five-piston stimulus set (five single whisker and 26 multi-whisker stimuli), thus covering the stimulus space, or if instead specific combinations were strongly over-represented. *Figure 3A* shows tuning curves of 31 neurons from one mouse that covered the stimulus space. Importantly, across all imaged mice, all 31 stimuli were represented approximately evenly (*Figure 3C*). To ensure the robustness of this analysis, we computed each neuron's preferred stimulus on 50% of the data and used it to order tuning matrices computed on the remaining 50% (see Materials and methods). This demonstrates that barrel cortex has the ability to sparsely encode all possible higher-order tactile stimuli in the five-whisker space we probed. Comparing the stimulus selectivity and deviation from linear summation from each neuron side by side shows that the population code for multi-whisker stimuli correlated well with selective supra-linear summation for the most preferred stimulus (*Figure 3B and D*). This supports the idea that stimulus-specific summation over surround whiskers helps promote a sparse and selective code for higher-order tactile features: for most whisker combinations, summation was sub-linear (i.e. suppressive, consistent with prior findings). However, for a small subset of multi-whisker inputs summation was strongly supra-linear. Another important rule emerged from this analysis: the best stimulus for each neuron nearly always contained the principal whisker (PW), demonstrating that it was required for the preferred supra-linear response (71% of neurons in C1 column, see Materials and methods, *Figure 3—figure supplement 1A, C*). The preferred stimulus also almost always included the anatomically aligned 'columnar' whisker (CW) even when the CW and PW were different (60% of C1 neurons' PW was C1, *Figure 3—figure supplement 1B*).

We next compared the population codes in L4 (2,428 neurons) and L2/3 (2,328 neurons): first, we found that sparse neural ensembles in both layers encoded the complete set of the 31 presented second-order tactile stimuli (*Figure 3—figure supplement 2A, B*), implying that even at the first stage of cortical processing cortical integration gives rise to a sparse, comprehensive code for higher-order stimuli. However, we observed several important differences indicative of a translaminar transformation of the neural codes for these features. In L4, the PW contributed substantially more to the individual neurons' responses than in L2/3. Conversely, in L2/3, the surround whiskers, often in highly specific and non-contiguous combinations, shaped the tuning to multi-whisker stimuli. Ultimately, as a consequence, a much greater fraction of L2/3 neurons responded more specifically to unique combinations of whiskers, than in L4. More than three times as many L2/3 neurons as in L4 exhibited multi-whisker receptive fields (*Figure 3—figure supplement 2C*), and L2/3 neurons were substantially less selective for single whisker stimuli (*Figure 3—figure supplement 2D, E*). Strikingly, L2/3 had nearly three times the number of neurons that exhibited driven responses for a multi-whisker stimulus that did not include the PW (14 vs. 43%; Extended data. *Figure 3—figure supplement 2F, G*) implying a more diverse code for complex tactile features. L2/3 also had a greater fraction of neurons whose preferred stimulus lacked the 'columnar whisker' (analysis restricted to the C1 column, *Figure 3—figure supplement 2H*). Similarly, among these C1 column neurons, the contribution of the C1 whisker to neuronal tuning curves was significantly more distributed than for L4 neurons (*Figure 3—figure supplement 2I*). These differences suggest that L2/3 could faithfully represent the combination of whiskers in a given stimulus with a more spatially compact population of neurons.

## A similar mechanism also generates a comprehensive population code for higher-order stimuli in the mouse visual cortex

If the synthesis of higher-order feature codes through stimulus-specific spatial summation is a widespread feature of cortical population activity, then we should be able to observe a similar phenomenon

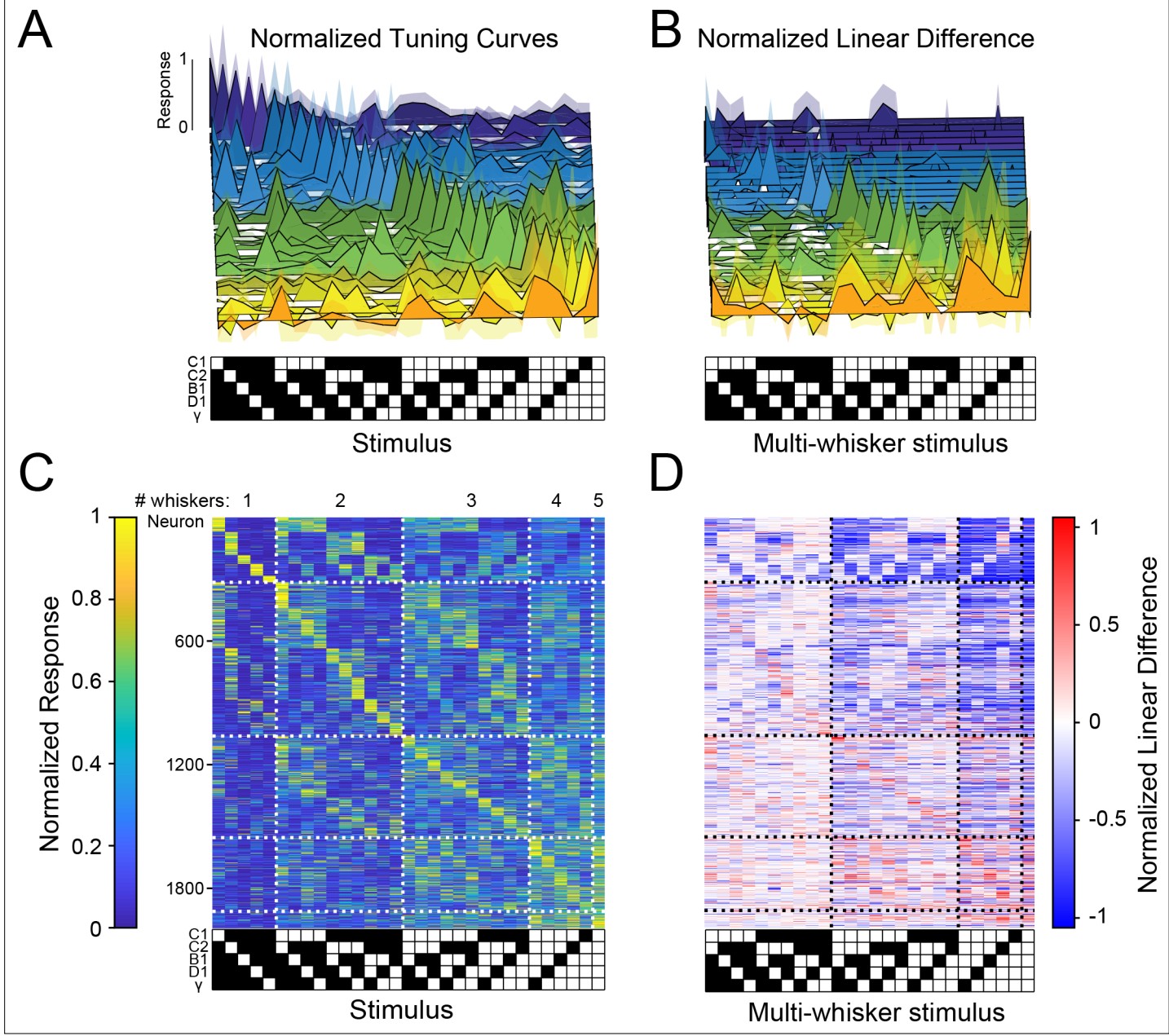

**Figure 3.** Sparse, high-dimensional population codes of higher-order tactile features generated through stimulus-specific supra-linear summation. (**A**) 31 example tuning curves from one field of view in one mouse showing neurons that exhibit a peak response for each of the 31 presented stimuli. Shaded regions are 95% confidence intervals (**B**) Linear difference of the 31 neurons in (**A**). (**C**) Normalized, cross-validated tuning curves for all sensory-driven neurons in awake, whisking mice. Neurons are ordered along the vertical axis by the stimulus to which they responded best. (**D**) The normalized linear difference for the neurons in (**C**).

The online version of this article includes the following figure supplement(s) for figure 3:

**Figure supplement 1.** The preferred multi-whisker stimulus for each neuron nearly always includes its principal whisker (PW).

**Figure supplement 2.** Differences between tactile representations in L4 and L2/3.

**Figure supplement 3.** Comparison between L4 and L2/3 of barrel cortex.

**Figure supplement 4.** Altered population codes in anesthetized, passively stimulated mice.

in a cortical area that receives extremely different sensory input, such as the primary visual cortex. To test this hypothesis, we made recordings of neural activity in L4 and L2/3 of mouse V1 while presenting five patches of drifting gratings, each approximately the size of the average classical receptive fields of most mouse V1 neurons, 10 visual degrees (see *Figure 4A* and Materials and methods). The drift direction of the center patch was orthogonal to those in the surround patches to ensure robust visual responses (*Cavanaugh et al., 2002*; *Self et al., 2014*; *Shushruth et al., 2013*). As in the barrel cortex, responses to single patch stimuli reflected the underlying topographic map (*Figure 4B*). In agreement with the notion of diverse, stimulus-specific summation, we likewise observed that V1 populations encoded all possible combinations of the five stimulus patches in approximately even fashion (*Figure 4C and E*). Moreover, just as in the barrel cortex, many V1 neurons summed these grating patches supra-linearly to give rise to their preferred stimulus, but summed nearly all other stimuli sub-linearly, effectively sharpening the tuning to the preferred stimulus (*Figure 4D and F*). As in barrel cortex (*Figure 3—figure supplement 3*), this coding principle was shared across layer 4 (*Figure 4—figure supplement 1A, B*) and layer 2/3 (*Figure 4—figure supplement 1C and D*). To ensure that this was not specific to the precise patch size of 10°, we also tested 15° patches, and observed qualitatively similar results (*Figure 4—figure supplement 2*).

To quantify the degree to which population codes reflected supra-linear versus sub-linear summation, we first sorted the stimuli in order of preference for each neuron (*Figure 5A*). Then, we computed linear differences as before for multi-whisker or multi-patch stimuli (*Figure 5B*). By averaging the linear difference for ranked stimuli across neurons (with ranking based on held-out data), we were able to quantify how supra- or sub-linearity of multi-whisker or multi-patch stimulus response varied with stimulus preference. We found that in multi-whisker or multi-patch preferring neurons across both cortical areas, sub-linear summation dominated for all but the most preferred stimuli (*Figure 5C and D*). In contrast, supra-linear summation was highly selective for only the most preferred stimulus on average across both cortical areas and layers. Across all layers and areas, the most preferred stimulus evoked significantly more supra-linear responses than the average non-preferred stimulus in a substantial fraction of neurons in both S1 and V1 (*Figure 5E and F*, *Figure 5—figure supplement 1*).

We next asked whether the supra- and sub-linear summation we observed could be captured by simple linear summation, followed by a static nonlinearity, as in a generalized linear model (GLM) (*Gerstner et al., 2014*). Using one-half of the trials (the 'training set'), we fit a seven-parameter GLM for each neuron. Across both layers and brain areas, these models captured tuning qualitatively well, for most neurons whose tuning curves we could estimate reliably (*Figure 5—figure supplement 2*, columns 1–3). We then compared performance of this model with that of an 'oracle model', in which the tuning is assumed to be perfectly captured by the measurements in the 'training set', (31 parameters, or one per stimulus condition). Indeed, we found that across layers and cortical areas, the GLM significantly outperformed the oracle model on average (*Figure 5—figure supplement 1*, column 4; $p < 10^{-4}$, Wilcoxon signed-rank test). Thus, we concluded that within the measurement error of our tuning curve estimates, a simple linear-nonlinear model was able to capture the stimulus-specific supra- and sub-linear summation we observed.

To ensure that our estimation of neuronal responses from calcium signals to the stimulus set is not substantially compromised by significant non-linearities in GCaMP6s, we measured the transformation from action potential rate to GCaMP6s fluorescence changes using two-photon targeted loose patch electrophysiology in *Camk2-tTa;tetO-GCaMP6s* mice (see Materials and methods, experiments were conducted in the visual cortex for technical reasons). We found this transformation to be approximately linear and highly sensitive to firing rate changes (mean Pearson's correlation coefficient=$0.71 \pm 0.03$, mean ± s.e.m., n=21), consistent with prior findings (*Chen et al., 2013*; *Figure 2—figure supplement 2*). For our barrel cortex data, we also tracked all five whiskers, including touches and bend, for all trials in the dataset, using an artificial neural network approach (DeepLabCut *Mathis et al., 2018*, see Materials and methods) to determine if sensorimotor adaptation in how the whiskers contacted the pistons could have explained the non-linear summation we observed in our neuronal responses. In a subset of mice (n=3) we measured five key parameters of the center whisker (C1): mean number of touches per trial, mean angle during the contact period, mean whisker bend, the amplitude of whisking, and the mean duration of contact. First, we compared these five kinematic variables between trials when individual pistons were presented alone with trials when each of the 26 combination stimuli were presented. However, we observed no difference in the number of touches, mean

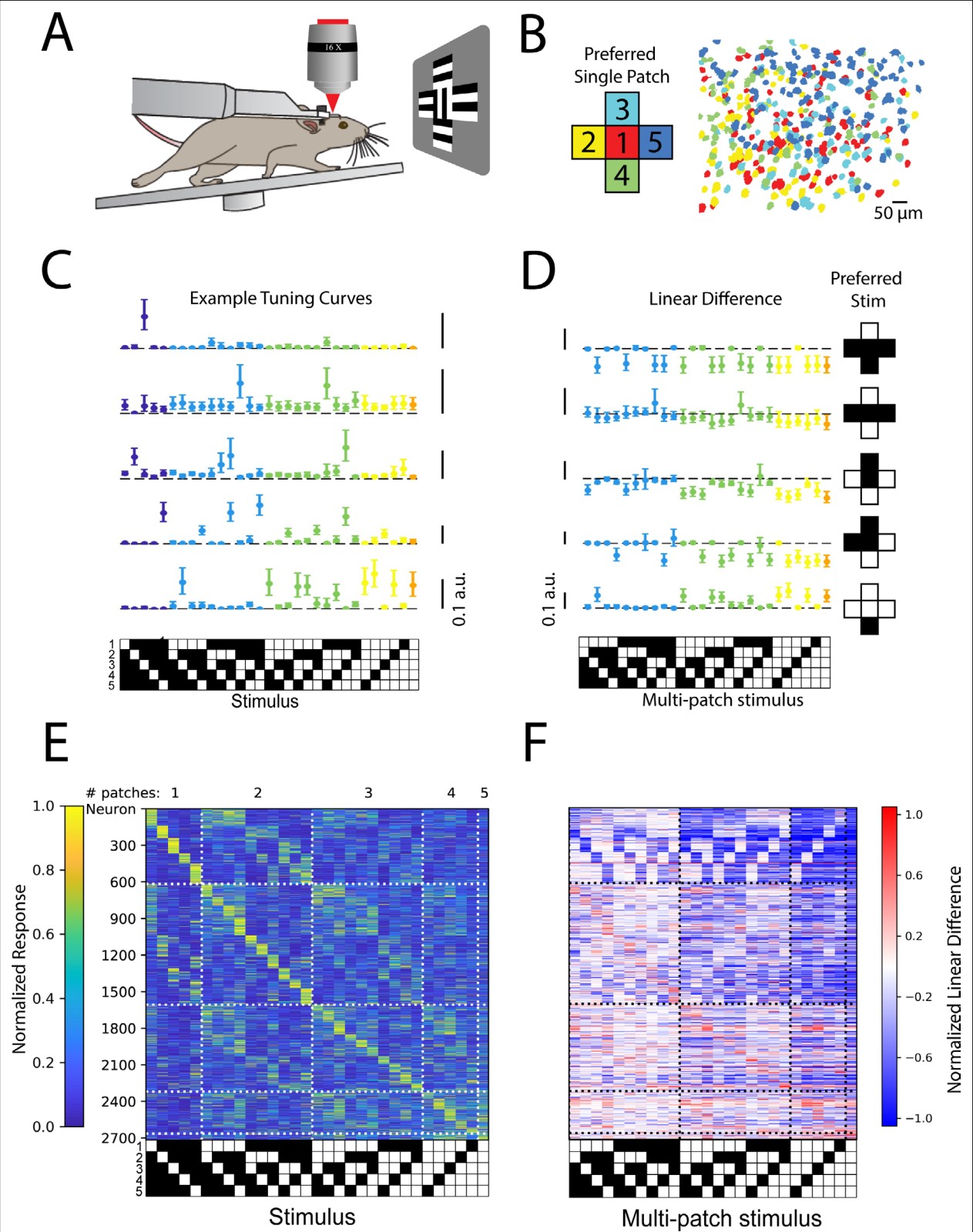

**Figure 4.** Stimulus-specific supra-linear summation in the primary visual cortex. (**A**) Schematic of the experimental preparation. (**B**) Example imaged field of view in layer 2/3 where each identified neuron is color coded according to its preferred single stimulus patch. (**C**) Five example tuning curves from neurons preferring different numbers of patches. Neurons are from four fields of view in layer 2/3 of three mice. Error bars are bootstrapped 95% confidence intervals (**D**) Linear difference of the five neurons in (**C**). (**E**) Normalized, cross-validated tuning curves for all sensory-driven neurons in

*Figure 4 continued on next page*

*Figure 4 continued*

awake, whisking mice. Neurons are ordered along the vertical axis by the stimulus to which they responded best. N=10 fields of view, 6 animals. (**F**) The normalized linear difference for the neurons in (**E**).

The online version of this article includes the following figure supplement(s) for figure 4:

**Figure supplement 1.** Comparison between L4 and L2/3 of visual cortex.

**Figure supplement 2.** Analogous population code for 15° grating patches.

**Figure supplement 3.** Effect of deconvolution on tuning curve calculation.

**Figure supplement 4.** Non-cross validated tuning curves and linear difference plots of S1 and V1 neurons.

**Figure supplement 5.** Alternative cross-validated tuning curves and linear difference plots of S1 and V1 neurons.

**Figure supplement 6.** Supra-linear difference plots reflect true deviation from linearity in S1.

**Figure supplement 7.** Eye movements are not strongly synchronized to stimulus presentation.

bend, or whisk amplitude, but did detect a decrease in mean contact duration and mean angle with the increasing number of pistons (see *Figure 2—figure supplement 3* and Materials and methods). These data imply that stimulus-specific behavioral adaptation in the mouse's whisking could not have contributed to the stimulus-specific supra-linear facilitation we observed, although it may have led to a general trend of modestly weaker sensory responses for stimuli containing an increasing number of pistons that would be systematic across all neurons.

Brain state and behavioral state can profoundly influence cortical computation. Whether they also affect how cortical neurons integrate over space to encode higher-order stimuli remains uncertain. In the barrel cortex, we compared spatial summation over the whisker array between the awake, active whisking state described above with passively stimulated conditions in anesthetized mice. In lightly anesthetized mice (to prevent active whisking) we stimulated the same five whiskers with a set of five piezo actuators (see Materials and methods). Although multiple previous studies have used noise stimuli (*Brumberg et al., 1996*; *Estebanez et al., 2012*; *Ramirez et al., 2014*) or specific deflections (*Higley and Contreras, 2005*; *Laboy-Juárez et al., 2019*; *Mirabella et al., 2001*) in anesthetized or sedated animals to estimate neuronal receptive fields and the impact of surround whiskers in S1, we used mechanical stimuli that approximated the movements of the whiskers during active sensation to more specifically comparing barrel cortex coding between active and passives states. Population responses in these anesthetized mice were not as uniformly distributed across multi-whisker space as in awake, whisking mice, and instead favored broader spatial summation, or in other words, exhibited poorer feature selectivity (*Figure 3—figure supplement 4A, C, E*). Furthermore, L2/3 tuning curves in awake mice were substantially less well fit by a general linear model, potentially owing to enhanced nonlinearity in the awake, active condition (*Figure 3—figure supplement 4F*). These results imply that anesthesia and passive touch linearize summation in L2/3 of the barrel cortex and alter the population code of higher-order features. Taken together, our data demonstrate that brain and behavioral state potently modulate how somatosensory populations respond to higher-order stimuli.

## Discussion

Our data show that a precise but heterogeneous logic of spatial summation in the sensory cortex leads to the generation of a sparse comprehensive feature code for complex tactile or visual stimuli. We found that the activity of barrel or visual cortical neurons, particularly in the awake, active state, could be potently facilitated by highly specific subsets of surround stimuli, and many neurons could only be driven by more complex spatial features. In each neuron, this surround facilitation was highly specific to just one or a small number of multi-whisker combinations, while most other combinations suppressed or had no impact on the response to preferred stimuli. Most importantly, across simultaneously imaged populations within a small region of the sensory cortex, the logic of surround summation was sufficiently diverse that ensembles of co-activated neurons could sparsely and uniquely encode any of the presented higher-order stimuli. Thus, a small, but relatively uniform fraction of the imaged neurons encoded each of the presented stimuli, and as a population, these neurons encoded a sparse, but a comprehensive representation of the stimulus space. This contrasts with an alternative, and perhaps simpler coding scheme that could also have given rise to a complete code for all stimuli: neurons could have acted as 'labeled line' encoders of individual whiskers, and populations of such

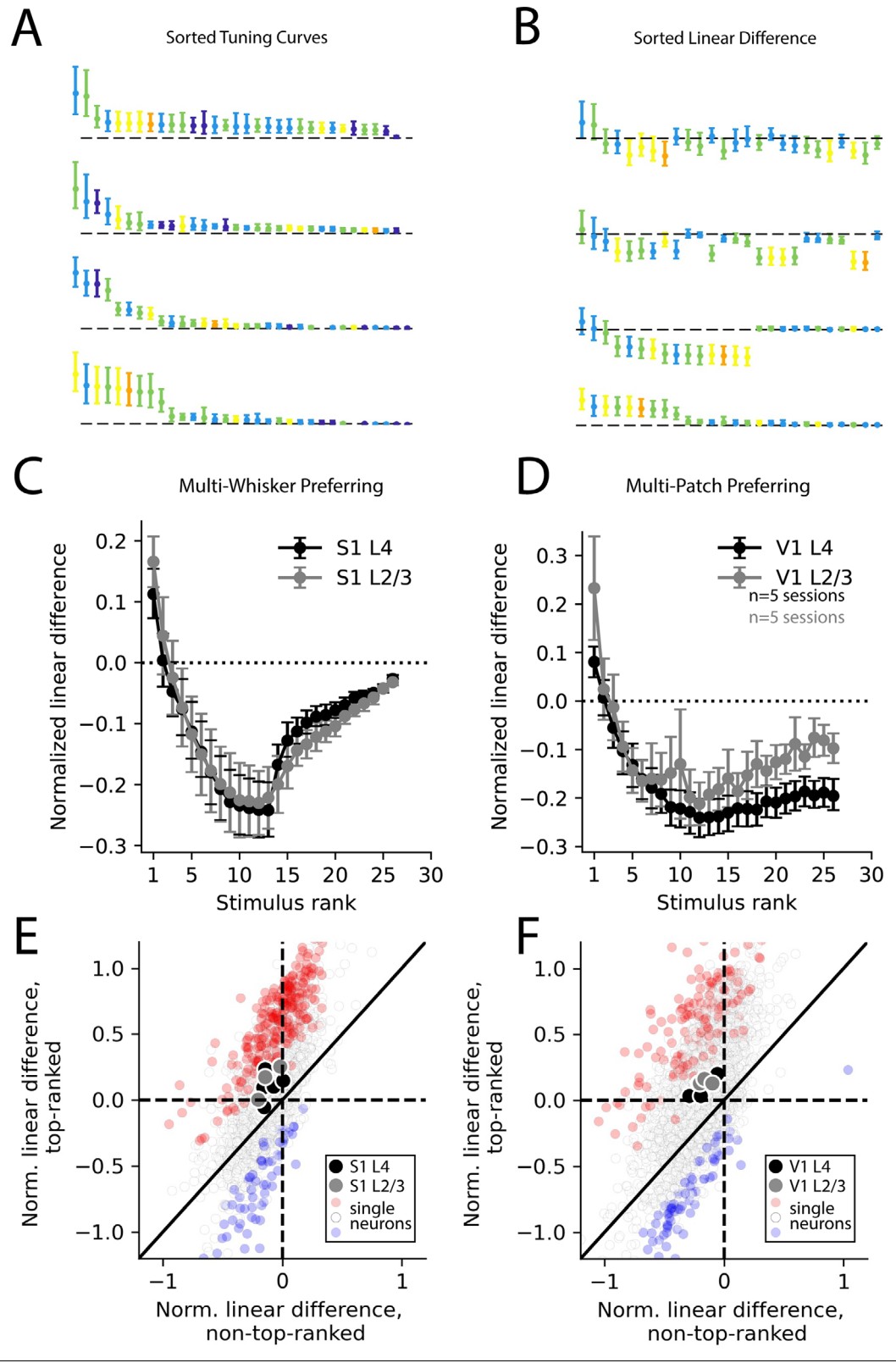

**Figure 5.** Selective supra-linearity and global sub-linearity of spatial summation across cortical areas and layers. (**A**) Example V1 layer 2/3 tuning curves of the four multi-patch preferring neurons reproduced from *Figure 4C*, sorted by descending response magnitude. (**B**) Linear difference of the same neurons' multi-patch stimulus responses, sorted as in (**A**). (**C**) Sorted linear difference averaged across neurons that preferred a multi-whisker stimulus in

*Figure 5 continued on next page*

*Figure 5 continued*

S1 L4 and L2/3. (**D**) Same as (**C**) for V1 L4 and L2/3. (**E**) Scatter plot of linear differences for preferred (top-ranked) stimulus versus an average of all non-preferred (non-top-ranked) stimuli, for individual neurons (small dots) and for averages across all neurons in an imaging session (large dots) for all S1 recordings. For single neurons, red color indicates significantly more supra-linear response to the top-ranked stimulus, compared to non-top-ranked stimuli, and blue color indicates significantly more sub-linear response (each p<0.01, using bootstrap resampling). (**F**) Same as (**E**), but for all V1 recordings. In all imaging sessions in both cortical areas, average supra-linearity is greater for the top-ranked stimulus than for non-top-ranked stimuli.

The online version of this article includes the following figure supplement(s) for figure 5:

**Figure supplement 1.** Supra-linearity of response to preferred stimulus reflects true deviation from linearity.

**Figure supplement 2.** Input-specific supra-linear integration is well captured by a simple linear-nonlinear transformation.

single-whisker selective encoders could have combinatorially encoded any of the stimuli presented in this study. The specific advantage of the more heterogenous scheme we observed remains to be further elucidated. One possible advantage could be to allow linear discrimination of arbitrary subsets of sensory input combinations, for learning complex classes and rules.

We note that the stimuli used in this study were not 'natural,' even though all data were collected in the active whisking state in awake mice. However, the precise control of the novel pneumatic stimulator allowed us to directly and quantitatively assess how individual neurons and large populations encoded multi-whisker stimuli during active whisking – something that has almost exclusively been done in the anesthetized or passive state which is arguably even less natural. Previous work in anesthetized rodents has shown how barrel cortex encodes global stimulus properties like moving wavefronts and tactile 'scenes' that would likely be generated by whisking at angled surfaces (*Drew and Feldman, 2007*; *Ego-Stengel et al., 2005*; *Jacob et al., 2008*). An important question is whether mice might ever encounter physical objects that could generate the complex multi-whisker touches we probed here. Although high-speed 3D imaging in animals freely whisking at natural objects would be needed to better answer this question (*Hobbs et al., 2015*), the complex 3D structure of natural surfaces could lead to moments when select sets of non-adjacent whiskers could come into contact with objects. Additionally, whiskers are of variable length and physical properties (*Yang et al., 2019*), which could also lead to non-intuitive spatio-temporal sequences of whisker-object contacts.

Our findings also raise the possibility that heterogeneous spatial summation of inputs beyond the receptive field might be a more general feature of cortical population coding, particularly in active contexts. Future work, perhaps combining functional connectomics (*Bock et al., 2011*) and multiphoton optogenetics (*Mardinly et al., 2018*; *Papagiakoumou et al., 2010*), could elucidate the precise circuit mechanisms that give rise to such a diversity of stimulus-specific supra-linear summation within a local population of neurons. In the mouse visual cortex, neurons with preferences for similar visual stimuli show a significantly greater tendency to synaptically couple (*Ko et al., 2011*). Moreover, subnetworks of pyramidal cells can preferentially interconnect based on their projection targets (*Brown and Hestrin, 2009*). Both of these connectivity motifs may lead to recurrent amplification of feature selectivity in specific subnetworks and could help explain the generation of the higher-order feature codes observed here, potentially further relying on the target specificity of horizontal projections (*Gilbert and Wiesel, 1983*; *Li and Gilbert, 2002*; *Petersen and Crochet, 2013*). Additionally, stimulus-specific modulations in sensory adaptation might contribute to the generation of feature selectivity, as is true in paralyzed rats, although this was only reported for neurons in deeper layers (*Ramirez et al., 2014*). Finally, feedback from higher cortical areas or secondary thalamic nuclei (*Jouhanneau et al., 2014*; *Williams and Holtmaat, 2019*), perhaps conveying signals related to self-motion or sensory predictions, might be critical for generating the diverse supra-linear response to specific complex stimuli, perhaps through non-linear activation of pyramidal cell dendrites (*Petreanu et al., 2012*; *Xu et al., 2012*). Such dendritic activity might be facilitated by local dendritic clustering of specific afferents (*Iacaruso et al., 2017*; *Takahashi et al., 2012*; *Wilson et al., 2016*), or circuits for dendritic disinhibition that are recruited by top-down motor-sensory projections (*Lee et al., 2013*). In addition to understanding its detailed circuit mechanisms, future work can address how higher-order feature selectivity in the primary sensory cortex gives rise to ever more selective representations for the identification of specific objects.

## Materials and methods

### Experimental model details

All experiments were performed on mice between 1.5 and 9 months of age. *Camk2*-tTA mice (RRID:IMSR_JAX:003010) crossed to tetO-GCaMP6s mice (RRID:IMSR_JAX:024742) were used when imaging L2/3. For S1 experiments, both lines had been outcrossed to the ICR line (Charles River) for several generations. Experiments imaging L4 in S1 used Scnn1a-Tg3-Cre mice (RRID:IMSR_JAX:009613) that had been outcrossed to the ICR line for several generations. For V1 L2/3 experiments, mice were on a mixed background between outcrossed tetO-GCaMP and camk2-tTA on the C57B/6 background. For V1 L4 experiments Scnn1a-Tg3-Cre mice were crossed to Ai162(TIT2L-GC6s-ICL-tTA2)-D mice (RRID:IMSR_JAX:031562). Both female and male animals were used and maintained on a 12:12 reversed light:dark cycle. All procedures were approved by the Animal Care and Use Committee of UC Berkeley.

### Preparation for in vivo two-photon imaging

Headplate attachment, habituation to running on a circular treadmill, intrinsic imaging, and cranial window installation were performed as described previously (*Pluta et al., 2017*). Briefly, anesthesia was induced with 5% isoflurane and maintained at 1–3% during surgery. Respiratory rate and response to toe/tail pinching were monitored throughout surgery to ensure adequate anesthetic depth; 0.05 mg/kg of buprenorphine was administered subcutaneously for post-operative analgesia. The scalp was disinfected with 70% alcohol and 5% iodine. The skin and fascia above the sensory cortices were removed and Vetbond (3 M) was applied to the skull surface and wound margins. A custom stainless steel headplate was fixed to the skull with dental cement (Metabond). For S1 mice, 2 days after surgery, mice were habituated to head-fixation on a free-spinning circular treadmill. This was repeated for 4–8 days until they freely ran at a fast and steady pace (>30 cm/s). Since mice often run in bouts, criterion was set such that mice had to maintain an average run speed above 10 cm/s over an hour to be included in the S1 component of the study.

Scnn1a-Tg3-Cre mice that reached running criterion were injected with AAV9-flexed-CAG-GCaMP6s (UPenn Vector Core) virus in left S1. Briefly, they were anesthetized and administered buprenorphine as described above. A dental drill (Foredom) was used to create a small bur hole 1.3 μm posterior and 3.5 μm lateral to bregma (marked previously with a sharpie during the headplate procedure). Then a WPI UltraMicroPump3 injector was used to inject 300 nL of the virus at a depth of 350 μm and a rate of 0.5 nL/s. Post-injection, the needle was left in the brain for 5 min to allow the viral solution to absorb into the tissue. Injected mice were provided 2–3 weeks with intermittent head-fixation over the circular treadmill to allow the infected neurons to ramp up the expression of GCaMP6s.

A cranial window was installed to provide for optical access to the cortex. The mice were anesthetized and administered buprenorphine as described above, and administered 2 mg/kg of dexamethasone as an anti-inflammatory. Post-dexamethasone injection, isoflurane was maintained at 1–1.5% during surgery. A dental drill was used to drill a 3 mm diameter craniotomy over the left primary somatosensory cortex. For V1 experiments, a biopsy punch was used to create a 3.5 mm diameter craniotomy over the left primary visual cortex. A window plug consisting of two 3 mm diameter coverslips glued to the bottom of a single 5 mm diameter coverslip using Norland Optical Adhesive #71 was placed over the craniotomy and sealed permanently using Orthojet. Mice were provided at least two days to recover.

For S1 experiments, intrinsic optical imaging was performed through the cranial window to localize the C1 and C2 barrels. Prior to imaging, anesthesia was induced as described above and then the mice were administered 5 mg/kg Xylazine. Anesthesia was maintained with 1% isoflurane during imaging. Imaging and stimulation were conducted using custom software written in (MATLAB). Mice were given 24 hr to recover.

### Tactile stimulus presentation and in vivo imaging of S1

Aluminum was custom machined to hold five pneumatic pistons on ball joints equally spaced along a circle. Pistons had a diameter of 1.6 mm, a length of 3 cm, and their tips were cut in such a way as to provide a flat surface for the whisker to palpate against. The apparatus was attached to a magnetic stand that could easily be slid along the optical table and locked in place.

Prior to the experiment, the mouse's whiskers were trimmed. Anesthesia was induced and maintained as described above. All whiskers were trimmed completely off, sparing the C2, C1, B1, D1, and Gamma whiskers. If one of these whiskers was missing, a nearby whisker was substituted (one mouse D2 for D1; one mouse beta for B1). The remaining five whiskers were trimmed in a staircase-like fashion such that the tip of the anterior whiskers in the C row wouldn't overlap with the tips of the posterior whiskers during whisking. Mice were head-fixed on a freely spinning running wheel under a Nixon ×16 magnification water immersion objective and imaged with a two-photon resonant scanning microscope (Neurolabware) within a light-tight box. Each piston was extended one at a time and adjusted such that its corresponding whisker contacted it close to the peak of the whisker's protraction and such that no other whisker could come close to coming into contact with it. A stereoscope and high-speed whisker tracking camera were used in positioning the piston and to verify contact was specific and repetitive. Calcium imaging occurred at 15.45 Hz with fields of view (FoVs) ranging from 800 μm by 1 mm to 1 mm by 1.3 mm. Wide-field reflectance imaging with a blue light-emitting diode (LED) was used to illuminate the vasculature and center the FoV on the region the intrinsic signal identified as corresponding to the C1 barrel. For L2/3 imaging, imaging depth was 100–300 μm, and for L4 imaging, depth was 350–500 μm deep. In some experiments, four depths 33 μm apart were imaged sequentially using an electrotunable lens (Optotune), with each depth sampled at an effective frame rate of 3.86 Hz.

Piston combinations were presented sequentially to the mouse's whiskers in a pseudo-random fashion where more weight was given to stimuli that had been presented the least. Stimulus presentation lasted 1 s followed by a 2-s inter-trial interval. Average running speed per trial was computed online, and trials, where the mouse ran slower than 10 cm/s were repeated later in the experiment. The experiment finished once each of the 32 stimulus combinations (including a catch/no stimulus condition) was presented 20 times while the mouse was running.

## Visual stimulus presentation, in vivo imaging of V1, and pupil tracking

For visual stimulus presentation, the monitor was placed 14–15 cm from the eye. Animals were habituated to visual stimulation on the setup for at least two sessions prior to imaging. The visual stimulus subtended one of five contiguous square patches laid out in a cross pattern, to match the spatial layout of pistons in the tactile stimulation experiments. Because contiguous homogeneous textures have been widely reported to drive sublinear responses, we focused on heterogeneous textures, with surround patches shifted in direction by 90° relative to the central patch. Trials with patches 10 visual degrees in diameter were interleaved with trials having patches 15° in diameter, and in some experiments, trials with 5° patches. To match the spatial scale of single S1 barrels, only the analysis of 10° patch presentation is presented in the main figures. Within each patch, square wave drifting gratings of 0 or 90° orientation were shown, with a spatial frequency of 0.08 cycles per degree, and a temporal frequency of 1 Hz. Stimulus presentation lasted one second followed by a one second inter-stimulus interval. Patch configurations, orientations, and sizes were pseudorandomly interleaved, and stimuli were generated and presented using the Psychophysics toolbox (*Brainard, 1997*). Each distinct visual stimulus was displayed for 10 repetitions.

For V1 imaging, the same Neurolabware setup was used as described above. The imaging FOV was 430 by 670 um, with four planes spaced 37.5 μm apart imaged sequentially, sampling each plane at an effective frame rate of 7.72 Hz. Electrical tape was applied between the objective and the mouse's headplate to block monitor light from entering the microscope.

Running was monitored in the same way as described above, with trials where maximum absolute run velocity >1 cm/sec classified as 'moving' and <1 cm/sec classified as 'non-moving'. Analysis was restricted to 'non-moving' trials.

Eye movements were imaged using a Basler Ace aCA1300-200um camera, with a hot mirror (Edmund Optics) placed between the eye and the monitor reflecting infrared light for eye imaging while transmitting visible light for visual stimulation. Infrared illumination was provided by the two-photon imaging laser, transmitted through the pupils, as well as a panel of 850 nm LEDs (CMVision). Pupil location and diameter were tracked using custom MATLAB code (*Figure 4—figure supplement 7*). When splitting the data into halves for cross-validated tuning curve and linear difference visualizations (see below), the halves were chosen to be balanced for pupil diameter and location.

## Anesthetized stimulus presentation

Preparation, trimming, and imaging conditions were consistent with above; however, this cohort of mice were not habituated to run while head-fixed. As well, immediately after whisker trimming the anesthetized mice, they were injected intraperitoneally with 0.075 mg chlorprothixene dissolved in saline. Ten minutes later, the isoflurane was turned down from 2to 0.5% and the mice were injected intraperitoneally with 40 mg urethane dissolved in saline. Mice were then head-fixed under the resonant scanning two-photon microscope, but this time over a feedback controlled heater (FHC). Up to 1% isoflurane was provided to maintain anesthetic depth, but was generally not necessary. A custom 3D printed whisker collar attached to a piezo plate bender (Noliac) was slid over each of the five whiskers. A cosine waveform designed to match the rate of angle change observed during natural touch was used to stimulate the whiskers (data not shown). The stimulus occurred at 16 Hz matching the average number of touches between the C1 whisker and its corresponding piston observed during one randomly selected experiment. Stimulus combinations were presented in a block randomized fashion, with each trial lasting 3 s, and the stimulus occurring for either 0.5 or 1 s. The experiment finished once each stimulus combination was presented 20 times.

## Calcium imaging analysis

For the S1 data, motion correction, region of interest (ROI) identification, and neuropil correction were calculated as described previously (*Pluta et al., 2017*). Briefly, two-photon movies were corrected for brain motion using Scanbox's sbxalign script (MATLAB; Mathworks) to correct for the rigid, 2D translation of individual frames. ROIs encompassing neurons were identified in a semi-automated manner using Scanbox's sbxsegmentflood (MATLAB) which computes and thresholds the pixel-wise cross-correlation for all pixels within a 60 by 60-pixel window. The ROI's signal ($R_i$) was taken as the mean value across all pixels within and unique to that ROI. This signal is assumed to be a mixture of the cell's actual fluorescence signal and a contaminating neuropil signal resulting from scattering producing off-target excitation, high illumination powers producing out of focus fluorescence, or unresolvable neurites passing through the microscope's point spread function. The neuropil signal ($N_i$) for each ROI was computed by averaging over an annulus of pixels surrounding the ROI but excluded pixels assigned to other ROIs as well as a smaller annulus of pixels that acted as a buffer in case any 2D motion artifact was not perfectly accounted for. This buffer annulus existed for all ROIs and was excluded from any neuropil calculation. As a result, the maximum diameter of the neuropil annulus varied per ROI in order to ensure a similar number of usable pixels to average over. Each neuron's neuropil-corrected fluorescence signal ($F_i$) was computed per ROI by the following equation:

$$F_i(t) = R_i(t) - k_i * N_i(t)$$

The amount of contamination ($k_i$) was assumed to be constant per ROI, but vary between ROIs as a result of local differences in expression and scattering. Each $k_i$ was defined by assuming that the neuron's true fluorescence signal ($F_i$) can never be negative (i.e. $k_i * N_i(t) \leq R_i(t)$), and that the neuropil signal cannot contaminate more than its measured value. The contamination coefficient per neuron was defined as follows:

$$k_i = \min\left(\frac{R_i(t)}{N_i(t)}\right); \text{if } k_i > 1, k_i = 1$$

A baseline fluorescence ($f_{0i}$) was calculated for each neuron by averaging its neuropil-corrected fluorescence over the last second of the inter-trial interval following a catch trial where no stimulus was presented, and then by averaging across the 20 catch trials presented. The change in fluorescence (ΔF/F) was calculated by subtracting off the baseline fluorescence from the neuropil-corrected fluorescence, and then dividing by the baseline fluorescence:

$$\Delta F/F = \frac{F_i(t) - f_{0i}}{f_{0i}}$$

Each neuron's response to a given trial was calculated as the mean ΔF/F over the 1-s stimulus period.

The C1 column was defined in a multistep process. First, a pixel-wise average across all running trials for each stimulus was performed, followed by averaging across the 1-s stimulus period to define the mean response to each stimulus for the whole field of view. The mean response to the no stimulus (catch) condition was then subtracted off from all averages. Next these averages were multiplied by the stimulus matrix to generate a linear model with a single weight for each whisker. Last each pixel was labeled by which whisker it responded the most strongly to creating a pixel-wise map of the preferred whisker. A Gaussian smoothing operation was used as well as dilation and erosion steps to generate smooth blobs defining the location of each column. The non-C1 columns were ignored as the C1 column is the only one whose borders were well defined since it's the only one whose surrounding whiskers were stimulated.

For the V1 data, motion correction and ROI segmentation were performed using Suite2p (*Pachitariu et al., 2017*). Neuropil subtraction was applied as described above. ΔF/F traces were calculated with baseline $F_0$ computed over a sliding 20th percentile filter of width 3,000 frames. Because the inter-stimulus interval was reduced in V1 recordings to permit more stimuli to be displayed, calcium transients overlapped between successive trials. Therefore, we deconvolved calcium traces for this data using OASIS with L1 sparsity penalty (*Friedrich et al., 2017*), using ΔF/F traces as input. We present a comparison of responses between deconvolved event rate and ΔF/F in *Figure 4—figure supplement 3*, showing a tight correlation. Deconvolved event rates were normalized such that a train of events producing calcium transients with time averaged ΔF/F=1, had a time-averaged magnitude of 1.

In order to ensure that deconvolution of calcium responses in the V1 data did not introduce biases into the calculation of tuning curves, we compared tuning curves computed with ΔF/F with tuning curves computed using deconvolved event rates. For this comparison, we examined trials following non-stimulus trials in the S1 data, in which stimuli were sufficiently widely spaced in time that calcium transients did not overlap between trials. In this data, we found that tuning curve values computed using the two methods were highly correlated (*Figure 4—figure supplement 3A*) with the majority (52%) of neurons showing Pearson correlation coefficients between deconvolved and ΔF/F tuning curves>0.9 in typical experiments (*Figure 4—figure supplement 3B*).

## High-speed whisker tracking

In all trials, the whiskers were imaged at high speed (300 frames per second) during the 1 s stimulus period using a camera (forward-looking infrared, or FLIR) with a telecentric lens (Edmund Optics) placed below the running wheel and reflected off a 45° mirror. The whiskers were illuminated in a trans-illumination fashion by a panel of 850 nm LEDs (CMVision) covered with a piece of tracing paper acting as a diffuser. The illumination source was placed at a sufficient distance as to create a flat background of uniform intensity.

Whiskers were later identified and tracked across time in a subset of experiments using DeepLabCut (*Mathis et al., 2018*). The neural network was trained on each mouse individually using ~160 frames spanning all piston conditions in the experiment. Frames in each condition were chosen automatically using k-means clustering (according to intensity values in pixel space; one frame was chosen from each cluster, k=5). In each frame, we labeled 30 points, 6 for each whisker (spread out uniformly along its length), that were used to fit the neural network. After visually evaluating the neural network's fit, the network was used to predict six points locating each whisker in each frame. Next, we fit second-degree polynomials to the network's points predicted for each whisker using the random sample consensus method (using random sample consensus, or RANSAC, with four minimum samples for the fitting). The whisker angle was estimated by fitting a line to the 10% of the whisker trace closest to the face and was defined relative to the frame's vertical axis. Outliers were discarded and replaced with the mean whisker angle from flanking frames. Subsequently, 95% of the whisker angle histogram was defined as the whisking range in a trial. The whisker bend ($\kappa$) was calculated on the third lower part of the whisker as $\frac{x'y''-x''y'}{\left(x'^2+y'^2\right)^{3/2}}$ where x and y are the x, y values along the whisker trace in the image's pixel coordinates. To estimate the overall force applied to the whisker, we subtracted the free-whisking internal curvature from the curvature values in the trial. Contact events were defined as when a whisker's trace overlapped with its respective piston's ROI. Each piston's ROI was defined by averaging together the frames in a trial where only that piston was present and then thresholding the image as the piston stands in stark contrast to the light background. Touch events for each whisker

were defined as periods of successive contact frames with separate events delineated by at least one frame without contact.

For each kinematic variable, we compared the multi-whisker trials (e.g. trials in which C1, C2, and D1 are stimulated) and the single-whisker trials (C1, C2, and D1 individually). Each parameter of the stimulated whiskers was averaged across all trials with a specific piston combination and averaged also within a trial (for the multi-whisker trials) or between trials (for the single-whisker trials). We compared each kinematic variable using paired student's t-test and corrected with the Benjamini & Hochberg procedure for multiple comparisons ($\alpha$=0.05). We further compared the kinematic variables for the C1 whisker in all conditions involving it using a one-way analysis of variance (ANOVA).

## Two-photon targeted loose patch calibration of GCaMP6s signaling

Adult *Camk2-tTa;tetO-GCaMP6s* mice were anesthetized with 2% isoflurane and kept warm with a FHC. After scalp and periosteum removal a drop of Vetbond was applied to the skull, and a small custom headplate was affixed to the skull with Metabond dental cement. A 3 mm circular craniotomy over left primary visual cortex was performed with a biopsy punch, and the brain was then covered with warm 1.2% agarose in phosphate-buffered saline (PBS). The dura was typically left intact. The mouse was injected with chlorprothixene (5 mg/kg) and urethane (1 g/kg). The mouse was mounted under a Sutter movable objective microscope two-photon microscope equipped with a 20 × 1.0 NA objective (Olympus) operated with ScanImage software (Vidrio). Anesthesia was maintained in a stable state with the addition of 0–1% isoflurane. Patch pipettes were filled with standard ACSF contained 50 µM alexafluor 594 K+salt (Thermofisher). The pipette was advanced under positive pressure and two-photon guidance through the dura/pia and towards GCaMP6s-expressing neurons in L2/3 at a 27° angle, carefully avoiding any blood vessels. During pipette advancement, full-field high-contrast drifting gratings were presented to generate neuronal responses in order to identify responding neurons by visual inspection. Cells with strong responses were targeted with the electrode in loose patch configuration using a Multiclamp 700B amplifier (Molecular Devices). Following acquisition of a low resistance seal, negative pressure was applied to increase the recorded amplitude of action potentials so that they were substantially larger than the background electrical noise, facilitating spike detection. The experiments then commenced. Trials were 3 s in length and consisted of a 500 ms baseline (gray screen) after which a high contrast grating drifted at 4–8 directions for 2.5 s, with 2 s intertrial intervals (gray screen). Visual stimuli were generated with Psychophysics toolbox (*Brainard, 1997*) and were displayed on 7″ LCD monitor (60 Hz refresh rate), positioned ~5–7 cm from the contralateral eye. All visual stimuli were full screen (~ 40° of visual angle) square wave drifting gratings with a temporal frequency of 2 Hz and a spatial frequency of 0.04 cycles per degree at 100% contrast. The LED backlight of the monitor was controlled by a high speed LED driver (Mightex) and triggered by the turn around signal from resonant scan mirror via a custom Arduino-based circuit so that the monitor was only illuminated during mirror flyback and not when the photomultiplier tubes (PMTs) were being sampled to avoid any optical artifacts from the monitor. A trigger from the visual stimulus computer synchronized the electrophysiology and two-photon imaging computers. Imaging power was 50–75 mW at the exit of the microscope objective. Electrophysiology data was acquired at 20 kHz and sampled via a National Instruments card (PCIe-6323) via custom MATLAB scripts. Spikes were extracted after digital high pass filtering (>1 kHz) and detected by a threshold set as samples greater than six times the standard deviation of the noise. Imaging data was analyzed as above following motion correction (which was minimal, but corrected nonetheless).

## Quantification and statistical analysis

Statistically significant differences between conditions were determined using standard parametric and nonparametric tests in MATLAB, including a one-way ANOVA, two-way ANOVA, two-sample t-test, Wilcoxon rank-sum test, and a Kolmogorov-Smirnov test. Analyses were performed on each ROI's measured ΔF/F for each trial. A single trial's response was calculated as the average ΔF/F during the entire 1 s of stimulation. Analysis was limited to ROIs that were significantly driven by at least one stimulus. A significant response for a stimulus had to meet two criteria: have a mean ΔF/F greater than 0, and pass a two-sample t-test between the evoked responses for a given stimulus and the measured ΔF/F values during catch trials. The Benjamini & Hochberg false discovery rate correction was used to correct for the multiple comparisons taken across the multiple stimuli. Outlier responses per stimulus

condition were identified by the median rule, where values further than 2.3 times the inter-quartile range from the median are determined to be outliers, and were removed prior to any analysis. Tuning curves were generated by averaging across all inlier trials, and 95% confidence were generated via bootstrap. In the awake mouse, these trials were further limited to running trials defined as trials where the mouse's run speed was above 10 cm/s and were not outliers as determined by again applying the median rule. Tuning curves subtracted off the mean response to catch trials to correct for any tuning offset.

The PW was determined by using Matlab's glmfit function to fit a general linear model to the individual trial responses as a function of what whiskers were stimulated. The model had one term for each whisker as well as an offset term.

A neuron was labeled as a single whisker neuron if it only exhibited a significant response to one of the five single whisker stimuli. It was labeled as a multi-whisker neuron if it responded significantly to two or more of the five single whisker stimuli. Or it was labeled as a combination neuron if it wasn't significantly driven by any single whisker stimuli, but was significantly driven by a multi-whisker stimulus.

For analyses comparing neurons' functional responses to their barrel column, barrel columns were first defined via a custom segmentation algorithm applied to a pixel-wise piston preference map. First, a pixel-wise average response was calculated for each piston by averaging over all trials in which that piston was presented. Then a 2D Gaussian smoothing filter was applied to each average image to reduce noise. Next, a pixel-wise preference map was created via labeling each pixel by the piston that produced the largest average response. Some pixels lie within resolvable neurons; however, most pixels lie between neurons and reflect the neuropil signal, an average of the surrounding population of neurons and fibers of passage. This preference map was then segmented into five discrete, non-overlapping blobs following an erosion, dilation, erosion, and Savitzky-Golay filter steps to ensure smooth, column-like edges (MATLAB). The segmentation process was iterated over to ensure the identified barrel columns were consistent with the known somatotopic map and architecture. Neurons were assigned to a barrel column if their centroid lied within an identified column.

In order to ensure the robustness of measured tuning properties, data for population-wide tuning curve and linear difference visualization was split into two halves, with trials balanced for stimulus conditions as well as pupil diameter and location in the case of the V1 data. Analysis was restricted to neurons showing a Pearson correlation coefficient >0.5 between tuning curves computed on the two halves of the data (i.e. neurons whose tuning curves could be reliably estimated based on half of the data). One-half of the data, the 'training set', was used to estimate stimulus preference, used for sorting the neurons. The other half of the data, the 'test set', was displayed. In this way, the 'preferred stimulus' was not mathematically constrained to evoke the largest response in the test set and could be taken as a robust feature of the data where this was the case. A similar procedure was used to produce the population-averaged sorted linear difference plots. Tuning plots for the full data set are presented in *Figure 4—figure supplement 4*. We additionally split the data into three thirds, balanced as before for stimulus conditions, and pupil location and diameter for the V1 data. This time, we restricted our analysis to neurons with a Pearson correlation coefficient >0.5 between tuning curves computed on thirds 1 and 2 of the data. We then used thirds 1 and 2 to estimate stimulus preference, and displayed third 3. In this way, we could ensure that no data from the third of trials used for display (third 3), were used for selecting neurons or estimating stimulus preference. Tuning curves and linear difference plots are presented in *Figure 4—figure supplement 5*.

To ensure that the procedure of sub-selecting reliably estimated tuning curves did not cause us to falsely detect supra-linearity based on noise, regardless of whether the data was split into halves or into thirds, we constructed surrogate datasets from the S1 data. In one set of surrogates, the underlying multi-whisker responses were linear sums of single-whisker responses. Independent noise was added to each partition (half or third, respectively) of the surrogate dataset, to match the standard deviation across partitions in the actual data. Unlike in the actual data (*Figure 4—figure supplement 6A, D*), a bright red diagonal did not appear in the normalized linear difference plots for this surrogate dataset (*Figure 4—figure supplement 6B, E*). We then constructed a second set of surrogate datasets, in which an identical stimulus-dependent signal was added to all partitions, with standard deviation matching the standard deviation across stimuli of the original tuning curve, as well as the same independent noise. Here, a bright red diagonal appeared (*Figure 4—figure supplement 6C*,

*F*). Similarly, the second set (with stimulus-dependent signal plus stimulus-independent noise), but not the first set of surrogate datasets (with only stimulus-independent noise) showed a substantial fraction of neurons with significantly stronger supra-linearity for preferred compared with non-preferred stimuli (*Figure 5—figure supplement 1*; when this second surrogate dataset was split into surrogate 'imaging sessions,' all but one showed greater supra-linearity for the preferred stimulus on average, similar to the actual data, in which all imaging sessions showed greater supra-linearity for the preferred stimulus). Thus, true deviation from linearity was necessary and sufficient for the appearance of supra-linearity in our analysis.

Sample sizes were not pre-computed, but chosen to be typical of similar studies in the field. Technical replications (referred to as fields of view, or imaging sessions in the text) were performed by imaging different areas of S1 or V1 on separate days in the same animals. Biological replications (referred to as animals in the text) were performed by imaging separate animals. One animal with one field of view was excluded due to having unstable receptive fields. This was determined by correlating each neurons' tuning from the second half of the experiment to that predicted by the first half. The excluded animal had a substantially larger fraction of neurons with correlation values less than 0.5. As well the excluded animal had much higher baseline fluorescence levels.

## Generalized linear model fitting

Trials were first split randomly into two groups, a 'training set' and a 'test set'. For each neuron's tuning curve computed on the training set trials, we fit a 7-parameter GLM (*Gerstner et al., 2014*) of the form

$$r_i = af\left(\sum_{j=1}^{5} w_j x_{ij} + h\right) + \varepsilon_i$$

where $r$ denotes the measured tuning curve, in deconvolved event rate, $i$ indexes over the 31 stimulus conditions, and $j$ indexes over the five whiskers (or visual stimulus patches). $x_{ij}$ is taken to be 0 if the piston (contrast) is not presented at whisker (patch) $j$ for stimulus $i$, and one if the piston (contrast) is presented. $h$ controls the baseline response, and $a$ controls the absolute amplitude of the response in units of event rate. $f$ is a static supra-linear function, defined as a unit standard deviation Gaussian convolved with a rectified linear function, as in *Miller and Troyer, 2002*

$$f\left(x\right) = \frac{x}{2}\left(1 + erf\left(\frac{x}{\sqrt{2}}\right)\right) + \frac{1}{\sqrt{2\pi}}exp\left(\frac{-x^2}{2}\right).$$

$\varepsilon_i$ is a Gaussian noise term, with identical standard deviation across stimuli $i$ (not fit). Maximum likelihood model parameters were estimated by minimizing the squared error between the modeled and measured training set tuning curve, using L-BFGS-B (*Byrd et al., 1995*).

We then quantified model performance as $R^2$ between the modeled and measured test set tuning curve. For comparison, we quantified the performance of an 'oracle model', with one parameter directly controlling the response to each stimulus condition (31 parameters total). In *Figure 5—figure supplement 1*, column 4, the difference between GLM and oracle test set $R^2$ is plotted for neurons with oracle test set $R^2 > 0.5$ .

## Electrophysiological recordings and analysis

Surgery preparation was similar as above but instead of implanting a cranial window the skull overlying the barrel cortex was thinned using a dental drill, and the C2 barrel region was localized by registering its intrinsic signal with the superficial vasculature. Next, a small cranial opening (<200 μm) was made to expose the C2 region of the barrel cortex. The exposed brain area was covered with silicone (Kwik-cast) and a subcutaneous injection of analgesic was administered before returning the mouse to its home cage. The following day, the animal was placed on the experimental rig to record neural activity during active touch.

During the experiment, while the mouse ran on the treadmill, its whiskers were imaged at 500 fps using an infrared camera (*Photonfocus DR1*). A laminar silicon probe (Neuronexus) was inserted approximately 850 μm deep into the C2 barrel using a micromanipulator (NewScale), and 32 analog signals were recorded onto a computer hard drive via an Intan 512 channel recorder (IntanTech). Using the same tactile 'display' as described above, single- and multi-whisker stimuli were pseudo-randomly

interleaved. Spikes were sorted offline using Kilosort 2.0 and manually curated using Phy. Whisker position in each video frame was determined using semi-supervised deep learning (DeepLabCuts). The onset of touch was determined using Matlab by identifying the time that each whisker entered the ROI of its corresponding touch surface. Histograms of spike rate were constructed using 5 ms bins.

## Acknowledgements

The authors acknowledge the GENIE Project, Janelia Farm Research Campus,and the Howard Hughes Medical Institute for the GCaMP6 viruses, as well as Dan Feldman and members of the Adesnik and Feldman labs for comments on the manuscript, and technical support from Kiarash Shamardani. HA is a New York Stem Cell Foundation-Robertson Investigator. DPM was supported by an NSF GRFP fellowship. This work was supported by The New York Stem Cell Foundation. This work was supported by NINDS grant DP2NS087725-01, NEI grant R01EY023756 and U19NS107613.

## Additional information

### Funding

| Funder | Grant reference number | Author |
| --- | --- | --- |
| NIH Office of the Director | DP2NS087725 | Hillel Adesnik |
| National Eye Institute | R01EY023756 | Hillel Adesnik |

The funders had no role in study design, data collection and interpretation, or the decision to submit the work for publication.

### Author contributions

Evan H Lyall, Conceptualization, Data curation, Formal analysis, Methodology, Software, Visualization, Writing – review and editing; Daniel P Mossing, Conceptualization, Formal analysis, Investigation, Methodology, Software, Validation, Writing - original draft, Writing – review and editing; Scott R Pluta, Conceptualization, Investigation, Methodology, Resources, Writing – review and editing; Yun Wen Chu, Investigation; Amir Dudai, Data curation, Formal analysis, Software, Visualization; Hillel Adesnik, Conceptualization, Funding acquisition, Project administration, Writing - original draft, Writing – review and editing

### Author ORCIDs

Evan H Lyall http://orcid.org/0000-0002-6946-7333
Daniel P Mossing http://orcid.org/0000-0002-9939-4788
Scott R Pluta http://orcid.org/0000-0002-3057-8095
Hillel Adesnik http://orcid.org/0000-0002-3796-8643

### Ethics

This study was performed in strict accordance with the recommendations in the Guide for the Care and Use of Laboratory Animals of the National Institutes of Health. All of the animals were handled according to approved institutional animal care and use committee (ACUC) protocols AUP-2014-10-6832-2 of the University of California, Berkeley. All surgery was performed under isoflurane anesthesia, and every effort was made to minimize suffering.

### Decision letter and Author response

Decision letter https://doi.org/10.7554/eLife.62687.sa1
Author response https://doi.org/10.7554/eLife.62687.sa2

## Additional files

### Supplementary files
• Transparent reporting form

## Data availability

All source data and analysis software is available on Dryad under https://doi.org/10.6078/D1370M.

The following dataset was generated:

| Author(s) | Year | Dataset title | Dataset URL | Database and Identifier |
|---|---|---|---|---|
| Mossing D, Lyall E, Adesnik H | 2021 | Data supporting: Synthesis of higher order feature codes through stimulus-specific supra-linear summation | https://doi.org/10.6078/D1370M | Dryad Digital Repository, 10.6078/D1370M |

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
