## [Editor Report]

This study provides now a comprehensive and well-documented proof that in both the visual and somato-sensory cortex the response to the sum of two stimuli is not the sum of the responses to individual stimuli in a large set of neurons. This is an important piece of work that further challenges linear models of cortical function and provide useful datasets to develop alternative non-linear models.

---

## [Decision Letter]

**Decision letter after peer review:**

Thank you for submitting your article "Synthesis of a comprehensive population code for contextual features in the awake sensory cortex" for consideration by *eLife*. Your article has been reviewed by 3 peer reviewers, including Brice Bathellier as Reviewing Editor and Reviewer #1, and the evaluation has been overseen by John Huguenard as the Senior Editor.

The reviewers have discussed the reviews with one another and the Reviewing Editor has drafted this decision to help you prepare a revised submission.

Summary:

How neurons integrate over spatial inputs to create complex receptive fields, the linear or nonlinear nature of this spatial summation, and the purposes it may serve, have been open questions in cortical sensory coding for several decades. They have remained open partly because of the complexity and high dimensionality of the parameter spaces that must be explored: e.g. does the nonlinearity depend on the identity of e.g. two whiskers in a pair being tested? on the time interval between their motions? on the orientation with which they move? This is one reason why it has been hard to come up with general governing principles. Periodically interest in the problem flares up, usually either because a complicated new nonlinear interaction is found or because an approach is proposed that will tame the complicated parameter space and provide a unifying framework. This manuscript describes cleverly designed experiments combining two-photon calcium imaging and whisker stimulation during head-fixed locomotion, to investigate the non-linearity of multiwisker input integration in barrel cortex. Based on comparisons between the responses observed for a single whisker contact and for multiple contacts authors aim to show that integration is not additive. Two-photon imaging in V1 similarly shows non additive integration. The approach is novel and has the advantage of probing the whisker system in naturalistic conditions, while ensuring a good control of the input stimulus. Overall this manuscript usefully contributes to the literature about non-linear responses in barrel cortex. It demonstrates that some important features of spatial integration (namely, a selective supralinearity that renders neurons highly preferentially sensitive only to specific combinations of whiskers or visual patches) are present in awake animals actively sensing their environment. This is potentially an important and interesting contribution and the paper merits consideration.

However some parts of the demonstration lack rigor. Potential analysis biases, although considered in the study, should be more carefully ruled out. Related to this, the description of analyses and of their rational could be more precise, the figure caption more detailed, and statistics could be strengthened. Also several points about the experimental design and interpretation need to be clarified.

1. Most of the demonstration is based on graphs in which responses are ranked according to the preferred stimulus. These graphs, as the authors state, are cross-validated to avoid confusion between the preferred response and a maximum resulting from response fluctuation. As shown in supplementary figure, without cross validation, integration looks extremely non linear and noise dominates the choice of the stimulus best driving a neuron. However, the authors do not provide a satisfactory solution to this problem. They choose to retain neurons with a correlation above 0.5 between responses for two distinct sets of trials. This of course help rejecting neurons in which noise dominates, but does not warrant the absence impact from noise in the data. So in principle there could still be noise-related biases which would corrupt the conclusions drawn from these graphs. To ensure an absence of biases: (i) one has to use a real cross-validation of the response maximum. A simple way is to rank maxima based on one half of trials, but display the responses averaged on the other half. In addition, simulations of mock dataset with the same noise and signal levels as the real data but in which all stimulus responses can be explained by a linear integration should be used as a null hypothesis, and an appropriate metric and statistics should be devised to prove that experimental observations are not explainable by this null-hypothesis model with a reasonable p-value.

Complementary information derived from population activity classifiers could be used to show that it is really possible to discriminate between the linear sum of individual responses and the actual mutiwhisker responses. At the very least, mixed effect statistical model (GLM) could be used to address the non-linearity of single neurons. That will allow to cross-validate the results with statistical control.

2. There's insufficient information about the whisker stimulation design for the reader to get a reliable sense of what/when are the whiskers actually touching, how this impacts on whisker motion, and what counts as a touch event for multiple whiskers. In the video provided (there should be more), whiskers often seem to be moving either "over" or "under" the piston so that even when they overlap with it in this projection, they seem not to be actually touching it. In other words, the whiskers seem to pass over the obstacle with no clear stop-go motion. Is motion tracking performed on the basis of just a 2D frame (this is never stated explicitly) or are there two frames to allow triangulation? This problem becomes compounded when trying to detect multiwhisker events. I think examples with much clearer evidence of touching need to be provided, and an explanation of how touches were detected and how this was checked. Similarly, in Figure 1 there seem to be only minor differences between moments marked as "touches" and others that aren't. At these traces' resolution, the only really clear instance of stop-go motion is the first gray area for single whisker C1. We need larger or higher resolution traces, or more examples, for clarity. Finally, coming back to multiwhisker touches: presumably these weren't perfectly simultaneous. How close together in time did the touches of multiple whiskers have to be, for them to be counted as a multiwhisker touch event?

3. Functional relevance of what's been shown. Why would there be any relevance to a comprehensive population code to report on punctual combinations of whiskers? This would be more understandable if the combinations of whiskers were being moved in a mutually coherent or ordered manner. Then one could test if neurons respond selectively to specific "modes" or "scenes" of collective stimulation, a plausible idea based on e.g. some of Shulz's or Feldman's work in anesthetized animals. As it stands, it seems really hard to give some kind of functional meaning to a neuron (e.g. Figure 2b/example 3 in Figure 2c,d) sensitive specifically to punctual stimulation of three non-adjacent whiskers. Under what conditions would those three whiskers tap simultaneously on an object (or objects)?? At least some discussion of this point would be useful.

4. There is an issue that the authors are clearly aware of: the use of calcium imaging methods to talk about facilitation and suppression. While the criticism of previous work with anesthetized animals showing surround suppression is valid, one can criticize this work for slow dynamics of calcium. When calculating the effect of a stimulus on the response with standard extracellular recording or whole cell recording, a PSTH typically involves 100ms to 500 ms around the stimulus. Stimulus evoked spikes are typically not evident 100 ms post stimulus in awake or anesthetized animals. In the whisker system, the bulk of stimulus evoked spiking is over in 50-100 ms, post stimulus. Here the responses are measured over 4 s – the comparison to studies with extracellular recording methods is fraught. Undoubtedly, the authors are correct that some neurons respond only to multi-whisker stimuli, and not at all to single whisker stimuli. And this has most likely been missed in previous work. But, how much facilitation would be seen if the same 4 s window was used with extracellular recording or targeted whole cell recording.

5. Another issue that applies to both visual and barrel cortex, is the clustering of single responses in layer 2/3 and 4. How does the clustering of tuning change from L4 to Layer 2/3. Looking at figure 2 and its supplement the layer 2/3 receptive field map shows no clear structure, and the layer 2/3 tuning shows no clear relationship to layer 4 tuning. One related issue that the authors seem to skirt around, is whether there is a spatial component to the multi whisker or multi-patch responses. Is it possible that location of neurons in a barrel, or nearer the edge of barrel explains part of the multiwhisker responses? Is it possible that neurons with facilitatory responses to patch stimuli are clustered?

6. One key and novel aspect of this work is that it is performed in the awake animal. This is a tremendous feat and should not be taken lightly. The stimulus presentation and tracking are all very well done. While it is all very nicely done, because it is in the awake animal, there are aspects of the presentation that seem to be incomplete. First it is not at all clear whether mice attend to the stimulus or that attention or brain state changes with stimulus condition. The authors should at least discuss that point or better specify the state of the animal. Second, awake means moving. So do mice move their eyes in relation to stimulus presentation? How do the dynamics of whisker movement affect the responses? It is highly unlikely that any two stimulus trials for whisker stimuli are identical -- the whisker that makes contact first will vary for each stimulus, the number of contacts will vary for each stimulus. The authors should better quantify this variability. Third in a whisker stimulus presentation for multiple whiskers the details of contact are not clear -- is the animal whisking to touch in all conditions? is the animal whisking differently for single versus multi-whisker stimulus condition? Do animals maintain contact with the all pistons, at the same time?

---

## [Author Response]

Revisions for this paper:1. Most of the demonstration is based on graphs in which responses are ranked according to the preferred stimulus. These graphs, as the authors state, are cross-validated to avoid confusion between the preferred response and a maximum resulting from response fluctuation. As shown in supplementary figure, without cross validation, integration looks extremely non linear and noise dominates the choice of the stimulus best driving a neuron. However, the authors do not provide a satisfactory solution to this problem. They choose to retain neurons with a correlation above 0.5 between responses for two distinct sets of trials. This of course help rejecting neurons in which noise dominates, but does not warrant the absence impact from noise in the data. So in principle there could still be noise-related biases which would corrupt the conclusions drawn from these graphs. To ensure an absence of biases: (i) one has to use a real cross-validation of the response maximum. A simple way is to rank maxima based on one half of trials, but display the responses averaged on the other half. In addition, simulations of mock dataset with the same noise and signal levels as the real data but in which all stimulus responses can be explained by a linear integration should be used as a null hypothesis, and an appropriate metric and statistics should be devised to prove that experimental observations are not explainable by this null-hypothesis model with a reasonable p-value.

This is a critical point. We have now included an additional supplementary figure (figure 4—figure supplement 5) in which we display the average response over trials not used in selecting neurons. To do this, we split the trials into thirds, and retain neurons with a correlation above 0.5 between the first and second third of the data. We then use the first two thirds together to determine the stimulus evoking the maximal response for each neuron. We plot the response averaged over the third third of trials. Responses displayed in this way appear qualitatively similar, showing stimulus-dependent supra-linearity.

Additionally, we have created a mock dataset, in which neural activity is taken to be linear sums of the single-whisker responses (which are as measured in the data), and Gaussian noise is separately added to each partition (third or half of the mock dataset), with across-partition variance matching the measured across-partition variance in the actual data (figure 4—figure supplement 6). Apparent stimulus-selective supra-linear responses are qualitatively absent in this analysis, whether splitting the data into thirds, as in figure 4—figure supplement 5, or into halves, as in the rest of the paper. A second mock dataset, in which an identical, random stimulus-dependent signal is added to all partitions, with across-stimulus variance matching the across-stimulus variance in the actual data, shows apparent stimulus-dependent supra-linearity. These analyses address the concerns of how noise influences the estimates of tuning and supralinearity.

Complementary information derived from population activity classifiers could be used to show that it is really possible to discriminate between the linear sum of individual responses and the actual mutiwhisker responses. At the very least, mixed effect statistical model (GLM) could be used to address the non-linearity of single neurons. That will allow to cross-validate the results with statistical control.

To address the mechanisms of nonlinearity in single neurons, we have now incorporated GLM fits (figure 5 —figure supplement 2), which show that a static supra-linearity applied to a linear sum of individual whisker inputs is sufficient to explain the tuning we observe, up to the precision allowed by our measurement error. We hope this adequately addresses the comment.

2. There's insufficient information about the whisker stimulation design for the reader to get a reliable sense of what/when are the whiskers actually touching, how this impacts on whisker motion, and what counts as a touch event for multiple whiskers. In the video provided (there should be more), whiskers often seem to be moving either "over" or "under" the piston so that even when they overlap with it in this projection, they seem not to be actually touching it. In other words, the whiskers seem to pass over the obstacle with no clear stop-go motion. Is motion tracking performed on the basis of just a 2D frame (this is never stated explicitly) or are there two frames to allow triangulation? This problem becomes compounded when trying to detect multiwhisker events. I think examples with much clearer evidence of touching need to be provided, and an explanation of how touches were detected and how this was checked. Similarly, in Figure 1 there seem to be only minor differences between moments marked as "touches" and others that aren't. At these traces' resolution, the only really clear instance of stop-go motion is the first gray area for single whisker C1. We need larger or higher resolution traces, or more examples, for clarity. Finally, coming back to multiwhisker touches: presumably these weren't perfectly simultaneous. How close together in time did the touches of multiple whiskers have to be, for them to be counted as a multiwhisker touch event?

We appreciate this critique and apologize for the ambiguity. We have edited the Results section, methods, and figure legends to clarify the nature of interaction between the stimuli and the whiskers and added an additional video with multiple trials. Briefly, all but five of the macrovibrissae were trimmed nearly completely. The remaining five whiskers were also trimmed so that all stimulus contacts would fit within the camera’s field of view and so that whiskers in a single row sweeping out the same azimuthal plane would only contact a single piston.

The stimulus device was set up with its five pistons in the ‘out’ position under a stereomicrocope to ensure that each of the five whiskers could only make contact with a single piston. All mice used in this study were selected based on their tendency to run at relatively high velocity while head-fixed for nearly the entirety of the experiment. Running at high velocity ensures that the whiskers are always actively whisking with a highly protracted set point (See Sofroniew et al. 2015, and Pluta, Lyall et all 2017). At time points when the animal was not running or running at lower velocity the whiskers would never contact the stimulus pistons and these trial were excluded. The stimulus device was setup when the mice were running at the same mean velocity as during the imaging so that their whiskers were in the approximately same position with the same setpoint.

The B2 and D2 whiskers were the only whiskers in their respective rows and thus could only contact the B or D row piston, unambiguously. Since the majority of whisker motion as other groups have shown is along the azimuthal axis (rather than the dorso-ventral axis), The B2 and D2 whisker never came close to touching a C row piston, and by the same token, the C row whiskers never came close to contacting the B or D pistons. For this reason, we only needed one camera with a moderately wide depth of field to capture all five whiskers, and any apparent crossing in the imaging data between the B2 and D2 whiskers with the C row pistons was a consequence of 2D imaging and not real contacts, and were thus excluded. The C-row whiskers were trimmed in a staircase manner with each posterior whisker being a little longer such that each stimulus piston placed close to the anterior tip of a whisker’s arc would only be contacted by that whisker. The staircase trimming meant more anterior whiskers were too short to touch that piston and the piston’s placement meant that more posterior whiskers never swept forward far enough to touch that piston. Thereby each whisker could only ever make contact with a single piston.

Motion tracking was performed on the basis of each 2D frame. The piston’s location in the extended position was segmented using a standard thresholding operation, while the whisker traces were identified using DeepLabCut. When the whisker trace overlapped with the piston ROI, it was registered as a contact event and touch events were registered as contiguous contact frames. Touches could be validated by the change in curvature of the whisker measured during such events. The methods section has been updated to clarify these points.

Our calcium imaging analyses were all performed trial-wise by analyzing the average change in fluorescence across the entire 1 second stimulus period, which encompasses around 10-20 whisk cycles. For the calcium imaging data, we do not make a determination of single vs multi-whisker touch events, rather we only disambiguate single vs multi-whisker stimuli. Unfortunately, our calcium imaging and the sensor itself (GCaMP6s) are too slow to correlate neural activity dynamics to underlying whisker kinematics. Repeating these experiments with large-scale neurophysiology could address this point in the future.

3. Functional relevance of what's been shown. Why would there be any relevance to a comprehensive population code to report on punctual combinations of whiskers? This would be more understandable if the combinations of whiskers were being moved in a mutually coherent or ordered manner. Then one could test if neurons respond selectively to specific "modes" or "scenes" of collective stimulation, a plausible idea based on e.g. some of Shulz's or Feldman's work in anesthetized animals. As it stands, it seems really hard to give some kind of functional meaning to a neuron (e.g. Figure 2b/example 3 in Figure 2c,d) sensitive specifically to punctual stimulation of three non-adjacent whiskers. Under what conditions would those three whiskers tap simultaneously on an object (or objects)?? At least some discussion of this point would be useful.

We agree and we have added to the discussion to address this concern. We appreciate that the stimulus paradigm we used was highly non-natural, although non-natural whisker stimulation has been extremely instructive throughout the investigation of the whisker system. It was designed, despite this fact, to allow precise measurements of combinatorial coding across whisker touches with different whiskers in the active whisking head-fixed state for the first-time. We cannot say for certain the coding scheme we discover here is indicative of how populations code in the natural environment. However, we can envision that when rodents whisk at complex 3D surfaces (rocks, the walls of underground tunnels, con-specifics, etc) their whiskers will generate somewhat unpredictable and heterogenous patterns of contacts with the whisker array. Natural objects have highly variable spatial structure so we could imagine cases where contacts across the array occur in complex and non-smooth patterns, although this warrant further investigations. Additionally, whiskers have non-uniform lengths adding the possibility that complex 3D objects would engage touch with non-adjacent whiskers.

4. There is an issue that the authors are clearly aware of: the use of calcium imaging methods to talk about facilitation and suppression. While the criticism of previous work with anesthetized animals showing surround suppression is valid, one can criticize this work for slow dynamics of calcium. When calculating the effect of a stimulus on the response with standard extracellular recording or whole cell recording, a PSTH typically involves 100ms to 500 ms around the stimulus. Stimulus evoked spikes are typically not evident 100 ms post stimulus in awake or anesthetized animals. In the whisker system, the bulk of stimulus evoked spiking is over in 50-100 ms, post stimulus. Here the responses are measured over 4 seconds -- the comparison to studies with extracellular recording methods is fraught. Undoubtedly, the authors are correct that some neurons respond only to multi-whisker stimuli, and not at all to single whisker stimuli. And this has most likely been missed in previous work. But, how much facilitation would be seen if the same 4 second window was used with extracellular recording or targeted whole cell recording.

Thank you for this point. Please note that when we quantified the calcium response on each trial it was due to multiple contacts on each trial (13.1 ± 5.9 contacts on average) occurring over 1 second, so the measured response was the integrated sum of spikes across all these contacts and was not restricted to a narrow 50-100 ms window and one contact. Note that the calcium responses were measured over only the 1 second stimulus period.

To address this point more directly, we now include extracellular electrophysiology from a group of mice in the identical stimulus paradigm. We add a supplemental figure (Figure 2 —figure supplement 4) with raster plots of spiking across several example units on a number of different trial types. First, the basic result we observe with the calcium imaging we also find in the electrophysiological data. Single neurons are highly selective to specific whisker combinations, and show supralinear summation to their component inputs. To ask how individual units respond to individual touches we generated PSTHs for individual units locked to contact times. This shows, as the reviewer expects, that the majority of spiking occurs within 40 ms of the contact, as seen in the added elecrophysiology figure. However, since many contacts occur between each whisker and its corresponding piston on each trial, when computing a PSTH across the entire trial one sees persistent spiking through the piston presentation time, consistent with the calcium imaging data (Figure 2 —figure supplement 4a,b). These electrophysiology data shows that we see essentially the same type of facilitation in spike rate as we do with calcium imaging (Figure 2 —figure supplement 4e), validating the results of the imaging data, and showing it was not a product of the slow temporal filtering of spikes by calcium sensing.

5. Another issue that applies to both visual and barrel cortex, is the clustering of single responses in layer 2/3 and 4. How does the clustering of tuning change from L4 to Layer 2/3. Looking at figure 2 and its supplement the layer 2/3 receptive field map shows no clear structure, and the layer 2/3 tuning shows no clear relationship to layer 4 tuning. One related issue that the authors seem to skirt around, is whether there is a spatial component to the multi whisker or multi-patch responses. Is it possible that location of neurons in a barrel, or nearer the edge of barrel explains part of the multiwhisker responses? Is it possible that neurons with facilitatory responses to patch stimuli are clustered?

Thank you for pointing this out. Indeed, we had analyzed the spatial responses in multiple ways but did not robustly uncover any specific structure reflecting multi-whisker preference, although we were limited by the sample size. However, in L4 we did find that neurons in a corresponding barrel nearly always have the corresponding whisker as their PW and this PW is a necessary component of their preferred multi-whisker stimulus. As to whether there is finer structure at the level of the location of neurons within each barrel, we prefer not to make any conclusions as we determined the barrel structure only using functional imaging which is probably not refined enough to determine the exact barrel borders. This is perhaps an even greater problem in L2/3 where the barrel boundaries are less clear. We believe this is an intriguing question for future work aimed specifically at addressing it.

6. One key and novel aspect of this work is that it is performed in the awake animal. This is a tremendous feat and should not be taken lightly. The stimulus presentation and tracking are all very well done. While it is all very nicely done, because it is in the awake animal, there are aspects of the presentation that seem to be incomplete. First it is not at all clear whether mice attend to the stimulus or that attention or brain state changes with stimulus condition. The authors should at least discuss that point or better specify the state of the animal.

We now add some text on the attention of the animals. We cannot say much for certain. The mice are not operantly trained – they are only habituated to run at high and consistent velocity while head-fixed. Thus, we don’t whether they were attending to the tactile stimulus or not – no reward was associated with the stimulus, but given the very strong physical interaction of the whiskers with the stimulator and the strong physiological responses in the brain, the stimulus is likely to be quite salient. Future work in which mice are trained to detect specific piston patterns could be highly informative on this question.

Second, awake means moving. So do mice move their eyes in relation to stimulus presentation?

During visual stimulation, there was little correlation between eye movements and the visual stimulus (at least in conditions where the stimulus predicts no reward). We have now included a supplemental figure (Figure 4—figure supplement 7) summarizing measurements of eye movements relative to trial timing in the V1 data. Eye movements tended to small and relatively infrequent. We plot the average motion across trials and it was ~0.02 degrees in the x and y dimensions. We did not track the eyes during whisker stimulation, so we cannot say for certain whether the eyes moved in some coherent fashion.

How do the dynamics of whisker movement affect the responses? It is highly unlikely that any two stimulus trials for whisker stimuli are identical -- the whisker that makes contact first will vary for each stimulus, the number of contacts will vary for each stimulus. The authors should better quantify this variability.

The reviewer is correct that each trial is unique with respect to the number, sequence, and duration of contacts across the whiskers. We capture the average response across 20 trials for each condition. From our whisker tracking analysis, we could quantify this variability. We now note this in the text. We agree that an analysis that captures response variability across 3D contact variability would be highly informative about population coding of multi-whisker contacts. However, as critiqued above, calcium imaging analysis is not suited to address this due to its slow kinetics. Instead, future work with electrophysiology would be highly suited to address this key question. We believe this is beyond the scope of this initial study.

Third in a whisker stimulus presentation for multiple whiskers the details of contact are not clear -- is the animal whisking to touch in all conditions? is the animal whisking differently for single versus multi-whisker stimulus condition? Do animals maintain contact with the all pistons, at the same time?

Yes, during whisker stimulation the animal is whisking in all conditions as stated in the Methods. Any trials without running, and hence without whisking, were excluded as the whiskers would not touch the stimulus device anyway as indicated in Methods. To address the latter question, we reported the data in Figure 2 Figure supplement 3 and noted this in the text (second to last paragraph). This figure tracks all of the five whiskers and compares various kinetic parameters between single piston stimuli (in this case the C2 piston) against the average of multi-piston stimuli containing the C2 piston. No difference was seen for the number of touches, bend, whisking range, or mean angle, but a small significant difference was observed for contact duration – with slightly shorter durations for stimuli with increasing number pistons. Yes animals can and often frequently contact all pistons at the same time, but also on some cycles only subsets of whiskers contact the pistons as the whiskers don’t always move coherently, as previously observed.